# Real-time classification of experience-related ensemble spiking patterns for closed-loop applications

Davide Ciliberti[1,2,3]\*, Frédéric Michon[1,2,3], Fabian Kloosterman[1,2,3,4]\*

[1]Neuro-Electronics Research Flanders, Leuven, Belgium; [2]Brain and Cognition, KU Leuven, Leuven, Belgium; [3]VIB, Leuven, Belgium; [4]imec, Leuven, Belgium

**Abstract** Communication in neural circuits across the cortex is thought to be mediated by spontaneous temporally organized patterns of population activity lasting ~50 –200 ms. Closed-loop manipulations have the unique power to reveal direct and causal links between such patterns and their contribution to cognition. Current brain–computer interfaces, however, are not designed to interpret multi-neuronal spiking patterns at the millisecond timescale. To bridge this gap, we developed a system for classifying ensemble patterns in a closed-loop setting and demonstrated its application in the online identification of hippocampal neuronal replay sequences in the rat. Our system decodes multi-neuronal patterns at 10 ms resolution, identifies within 50 ms experience-related patterns with over 70% sensitivity and specificity, and classifies their content with 95% accuracy. This technology scales to high-count electrode arrays and will help to shed new light on the contribution of internally generated neural activity to coordinated neural assembly interactions and cognition.

DOI: https://doi.org/10.7554/eLife.36275.001

\*For correspondence:
davide.ciliberti@nerf.be (DC);
fabian.kloosterman@nerf.be (FK)

**Competing interests:** The authors declare that no competing interests exist.

## Introduction

Distributed and temporally organized patterns of spiking activity in cortical circuits reflect the processing of sensory information, the execution of cognitive tasks and the generation of motor commands. Large-scale recordings have revealed the presence of brief (50–200 ms) multi-neuronal spiking sequences in cortical networks. The brief sequences appear as rapid increases in the multi-unit firing rate and they are generated endogenously or in response to an external stimulus (*Luczak et al., 2009Luczak et al., 2007*; *Luczak et al., 2009*; *Pasquale et al., 2017*; *Pastalkova et al., 2008*; *Riehle et al., 1997*). The omnipresence of these activity patterns in the cortical mantle suggests that they constitute the basic elements of the cortical neural code (*Harris, 2005*; *Luczak, 2015*).

Multi-neuronal sequential firing patterns have been extensively studied in the hippocampus, in large part because of the comprehensive characterization of a well-defined spatial code (*O'Keefe and Nadel, 1978*). In rodents, large-scale recordings revealed the occurrence of accelerated 'replays' of the spatial information encoded in the population activity during an experience (e.g. traversal of an arm in a maze), both during sleeping and when in a resting state (*Davidson et al., 2009*; *Diba and Buzsáki, 2007*; *Foster and Wilson, 2006*; *Karlsson and Frank, 2009*; *Lee and Wilson, 2002*; *Wilson and McNaughton, 1994*). Structured firing of experience-related activity was also reported in entorhinal (*O'Neill et al., 2017*; *Ólafsdóttir et al., 2016*), neocortical (*Hoffman and McNaughton, 2002*; *Ji and Wilson, 2007*; *Peyrache et al., 2009*; *Qin et al., 1997*; *Ribeiro et al., 2004*; *Rothschild et al., 2017*) and subcortical (*Girardeau et al., 2017*; *Gomperts et al., 2015*; *Lansink et al., 2008*; *Pennartz et al., 2004*; *Valdés et al., 2015*) circuits.

Correlative analyses of the multi-neuronal spike content of hippocampal replay suggest that distinct replay sequences might be used for sharing information with the neocortex, specifically for system-level memory consolidation (*Axmacher et al., 2009*; *Dupret et al., 2010*; *Lewis and Durrant, 2011*) and for the creation and update of an internal model for covert evaluation of possible outcomes during memory-guided decision-making (*Ólafsdóttir et al., 2018*; *Ólafsdóttir et al., 2017*; *Penny et al., 2013*; *Pezzulo et al., 2014*; *Pezzulo et al., 2017*;*Pezzulo et al., 2014*; *Pfeiffer and Foster, 2013*;*Wu et al., 2017*). Closed-loop manipulations of replay-related activity are to date limited to the use of real-time detection of fast sharp-wave ripple (SWR) oscillations (120 – 250 Hz) (*Ego-Stengel and Wilson, 2010*; *Girardeau et al., 2009*; *Girardeau et al., 2014*; *Jadhav et al., 2012*; *Kovács et al., 2016*; *Maingret et al., 2016*;*Nokia et al., 2010*; *Nokia et al., 2012*; *Novitskaya et al., 2016*; *Roux et al., 2017*; *Talakoub et al., 2016*; *van de Ven et al., 2016*) that reflect merely the bursting state of the neuronal population but not the actual replay content. Any causal link between replay content and behavior or physiology thus remains untested.

To provide a direct causal link between a specific transient neural activity pattern and its hypothesized contribution to cognition, correlational and computational studies are not sufficient and a closed-loop manipulation is necessary (*Hady, 2016*; *Potter et al., 2014*; *Zrenner et al., 2016*). A critical element for 'closing the loop' around a neural pattern of interest is the ability to detect such pattern in real-time (*Grosenick et al., 2015*), but a complete methodology for detecting (re) expressed spike patterns within tens of milliseconds has not yet been demonstrated.

The rapid detection of multi-neuronal patterns in a closed-loop setting is challenging because offline identification algorithms generally require all data to be available or are otherwise not suitable for an online scenario. Moreover, the algorithms often have intensive computational demands that pose challenges for real-time execution at tens of millisecond latency. Such requirements are not present in brain–computer interfaces (BCIs) and other neuroprostheses for motor control as they generally operate sufficiently with latencies of 120 – 200 ms (*Ibáñez et al., 2014*; *Velliste et al., 2008*; *Xu et al., 2014*), one order of magnitude greater than is required for the rapid detection of transient cortical multi-neuronal sequences.

To bridge this gap, we devised a system for online spike pattern detection with minimal computational time delays. The system builds on our previous work describing an efficient offline implementation of spike-sorting-less neural decoding (*Kloosterman et al., 2014*; *Sodkomkham et al., 2016*) and a general software framework (Falcon) for complex online neural processing in closed-loop settings (*Ciliberti and Kloosterman, 2017*). Here, we added parallelized neural decoding to Falcon and built on top a concrete implementation of a full real-time spike pattern identification and classification system.

Using this novel closed-loop system, we performed real-time population decoding of spatial information from CA1 assemblies at the stringent 10 ms temporal resolution and detected distinct classes of replay at a low 50 ms average latency. Real-time identification of population bursts with replay had more than 70% average sensitivity and specificity, whereas content classification accuracy was on average over 95%. We demonstrated this technology in a freely behaving rat during both resting and awake states using in vivo large-scale recordings. We also illustrated that detection algorithm parameters can be adjusted, depending on the requirements of the experiment, to trade off sensitivity and specificity or latency and accuracy. We further showed how detection performance is affected by the number of recording channels.

Our ready-to-use brain–computer interface will enable, for the first time, selective manipulation of specific hippocampal and extra-hippocampal ensemble firing sequences at a millisecond timescale and will thus pave the way towards a deeper understanding of their contribution to cognitive processes.

## Results

We built a fully functional closed-loop system for real-time detection and identification of replay patterns. Our system addressed three general major technical challenges. The first challenge derives from the desire to respond to an identified replay as soon as possible after the event has started (and well before it is over). Thus, the determination of replay content should be completed with minimal data, which inevitably introduces a trade-off between algorithmic detection latency and detection accuracy. Second, any approach for online detection of hippocampal replay content must be

suitable for data arriving as a continuous live stream; in such a scenario, the exact start (and end) times of replay events are unknown and cannot be predicted. The constraints of the online scenario require the implementation to make use of past-observed information only. As a third challenge, the real-time computations needed to identify replay content in the continuous high-volume and high-rate stream should only minimally increase the overall detection latency, which should be dominated by its algorithmic contribution. The need for fast computations introduces another possible latency–accuracy trade-off, in which approximate algorithms may be required to meet real-time deadlines.

Built on top of the Falcon open source framework for closed-loop neuroscience (*Ciliberti and Kloosterman, 2017*), the system reads and processes a continuous stream of multi-channel neural signals, analyzes the signals to search for potential replay events and emits a digital output pulse upon the detection of a replay event containing target content (*Figure 1a*, *Figure 1—figure supplement 1a*). We defined the target as a replay trajectory of one of the tracks in a multi-arm maze (e.g. a Y-maze). To detect replay content, we applied neural population decoding followed by the characterization of the spatial-temporal structure in the decoded locations (*Figure 1b*), similar to the approach that is often used for offline replay analysis (*Davidson et al., 2009*; *Kloosterman, 2011*). The incoming data stream is continuously decoded in 10 ms bins and, on the basis of the most recent 30 ms of decoded spiking activity, an assessment is made of whether or not a population burst with replay of one of the maze arms is present.

In the online replay classification algorithm, burst and content detections are combined (*Figure 1c*, *Figure 1—figure supplement 1b*, see 'Materials and methods'). The burst detection component responds to the crossing of a user-defined threshold ($\theta_{mua}$) in the causal moving average of multi-unit activity (MUA). The content detection component searches for sharp high-fidelity position estimates that are consistently located in the same maze arm in the last three 10-ms time bins. The sharpness of the estimates is computed as the causal moving average of the locally integrated probability around the maximum-a-posteriori (MAP) estimate and needs to cross a user-defined threshold ($\theta_{sharp}$). A replay event is identified if the burst, sharpness and consistency conditions are all met simultaneously.

The algorithms for neural decoding and replay content detection were adapted to fit the specific constraints of the online use case. In our approach, the most demanding computation is the online evaluation of the encoding model that directly relates spike waveform features to the animal's position without the need for prior spike sorting (*Kloosterman et al., 2014*). Modest compression of the encoding model reduced the computational costs, resulting in an online decode time below 1 ms per spike, without affecting the system's ability to decode the animal's position during run epochs (*Figure 1c*). The effect of compression on the integrity of posterior probability distributions in candidate replay population bursts is small for low to moderate compression levels, in particular for posteriors that have a MAP estimate with high probability (*Figure 1d*).

Overall, the implementation of the algorithms, including the spike-by-spike construction of posterior probability distributions and parallel processing (see 'Materials and methods'), made sure that the determination of replay content and the triggering of a corresponding digital output pulse occurred within 2.2 ms of the time at which the neural data were acquired in over 95% of the cases (*Figure 1e*; median added latency: 1.1 ms). Stress tests of the system with artificially generated spikes showed that the added latency remains low for high per-tetrode spike rates and for a large number of tetrodes (up to 32) on a 32-core workstation (*Figure 1—figure supplement 2a*), thus demonstrating the scalability of our solution. Additional tests on a 4-core workstation revealed that added latency has higher variability, especially in the case of 32 tetrodes and high spike rates (*Figure 1—figure supplement 2a*), but that performance is similar to that of the 32-core workstation for up to 24 tetrodes and per-tetrode spike rates that match our datasets (*Figure 1—figure supplement 2b*).

## Real-time detection of replay content during resting and awake states

To test our closed-loop framework of online detection of replay content, we recorded hippocampal ensemble activity using tetrodes in a rat that repeatedly explored the arms of a 3-arm radial maze. In each of the three separate recording sessions, a first exposure to the maze (RUN1) was used to prepare the encoding model for online neural decoding. Cross-validation showed good decoding performance in RUN1 with a median error below 10 cm (*Figure 1—figure supplement 3a*), which required only a few traversals of the maze (*Figure 1—figure supplement 4*). Real-time closed-loop

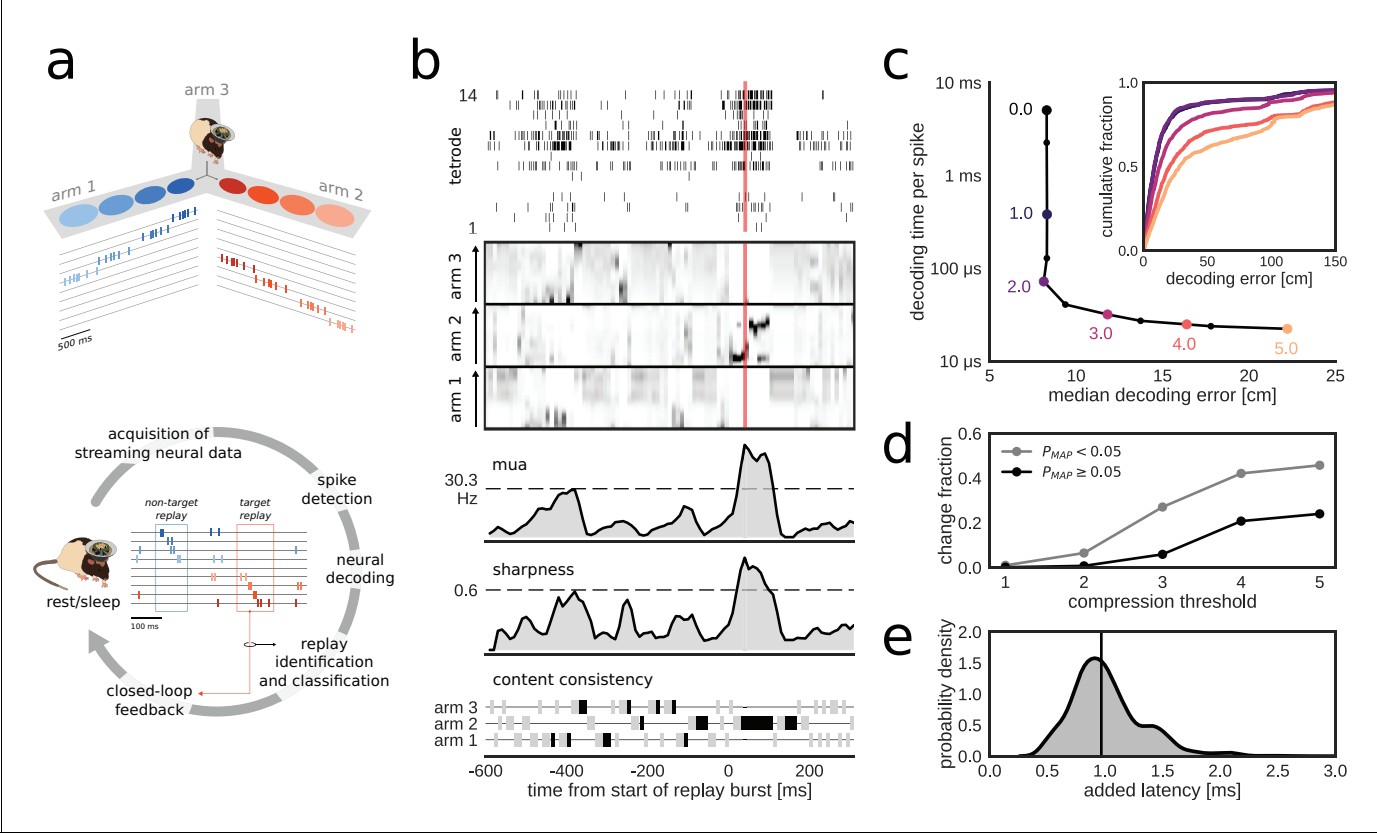

**Figure 1.** Online decoding and detection of hippocampal replay content for closed-loop scenarios. (**a**) Schematic overview of a system for online identification and classification of hippocampal replay content. Top: during the initial active exploration of a 3-arm maze, the relation between the animal's location and hippocampal population spiking activity is captured in an encoding model for subsequent online neural decoding. Bottom: streaming neural data are processed and analyzed in real-time to detect replay events with user-defined target content. Upon positive detection of a target, closed-loop feedback is provided to the animal within the lifetime of the replay event. (**b**) Algorithm for online classification of replay content. The top two plots show a raster of all spikes detected on a 14-tetrode array and the corresponding posterior probability distributions for location in the maze generated for non-overlapping 10-ms bins (gray scale maps to 0–0.1 probability). The lower three plots show the internal variables used by the online algorithm to identify and classify replay content. In the bottom plot, gray/black ticks indicate the lack/presence of content consistency across three consecutive bins. Red line: detection triggered when all variables reach preset thresholds (dashed lines for multi-unit activity (MUA) and sharpness, black tick for content consistency). (**c**) Effect of varying levels of compression on decoding performance and computing time. For the test dataset, compression thresholds in the range [1, 2.5] reduce the decode time to less than 1 ms per spike without affecting the accuracy of decoding the animal's location in 200 ms time bins during active exploration of a 3-arm maze. Decoding performance was assessed using a 2-fold cross-validation procedure. Inset: full cumulative distributions of decoding error for varying compression thresholds. (**d**) Effect of compression on posterior probability distributions computed for 10-ms bins in candidate replay bursts. For each level of compression, the plot shows the fraction of posteriors for which the maximum-a-posteriori (MAP) estimate changes to a maze arm different from the no-compression reference condition. Modest compression levels (≤2) have only a small effect on the MAP estimate, in particular for the most informative posterior distributions with MAP probability > 0.05. (**e**) Distribution of added latency, that is the time between the acquisition of the most recent 10 ms of raw data and the availability of the posterior probability distribution computed on the same data. Black vertical line represents median latency. Note that online spike extraction, neural decoding and online replay classification add a lag of less than 2 ms.

DOI: https://doi.org/10.7554/eLife.36275.002

The following figure supplements are available for figure 1:

**Figure supplement 1.** Neural decoding and replay identification over streaming data.
DOI: https://doi.org/10.7554/eLife.36275.003
**Figure supplement 2.** Added latency remains low with higher channel counts and spike rates.
DOI: https://doi.org/10.7554/eLife.36275.004
**Figure supplement 3.** Run decoding performance.
DOI: https://doi.org/10.7554/eLife.36275.005
**Figure supplement 4.** Decoding performance during exploration in all three datasets as a function of the training time and corresponding number of laps used to build the encoding model.
DOI: https://doi.org/10.7554/eLife.36275.006

replay content detection was activated in a subsequent rest epoch (REST) and in a second exposure to the radial maze (RUN2). The encoding model built using data from RUN1 was also valid for decoding the animal's position in RUN2 (*Figure 1—figure supplement 3b*), indicating that recordings were stable and spatial representations had not remapped (*Anderson and Jeffery, 2003*; *Latuske et al., 2017*). To evaluate the online detection performance without distortions of the signal, no actual feedback perturbation of brain activity was applied to the brain.

Reference hippocampal population bursts were defined offline on the basis of the detrended MUA signal and classified as either associated or not associated with replay (*Table 1*). The offline reference data set contained a total of 3506 population bursts that were recorded in REST (n = 1321) and RUN2 (n = 2185) across all three sessions. The median duration of reference bursts was 83.0 ms (inter-quartile range (iqr) [65.3, 119.8]; REST: median 89.0 ms, iqr [70.0, 126.0]; RUN2: median 79.0 ms, iqr [63.0, 116.0]). In RUN2, the majority of reference bursts occurred when the animal paused at the distal reward platforms.

Online, closed-loop replay identification resulted in a total of 2123 positive hits, a large majority of which (87.3%, 1854/2123) were associated with reference bursts. Very few online detections occurred immediately before and within 20 ms after reference burst onset, or after burst offset (*Figure 2a*). A small fraction of hits were non-burst false positives ($FP_{non\_burst}$), predominantly in the third test session (*Table 2*). There was a slightly higher fraction of positive hits inside reference bursts in REST (91.9%, 694/755) as compared to RUN2 (84.8%, 1160/1368). The higher rate of $FP_{non\_burst}$ in RUN2 is possibly due to the more prominent spatially modulated firing present during active exploration behavior, which led to spurious crossings of the $\theta_{mua}$ threshold with posterior densities representing the actual, rather than a replayed, position of the animal. Indeed, 80.6% of $FP_{non\_burst}$ in RUN2 occurred when the rat was moving (speed > 5 cm/s), whereas the majority of online detections inside reference bursts (73.8%) occurred when the rat was immobile (speed < 5 cm/s). Overall, the majority of online detections corresponded to population bursts.

If positive online hits that were associated with reference bursts correctly indicated replay content, then it is expected that, as compared to the ignored bursts, the detected bursts scored higher on metrics used for offline replay identification. Indeed, reference population bursts that were identified online as replay events had higher posterior bias ($bias_{max}$ and $bias_{max}$ score) for one of the maze arms than did reference bursts that were ignored by the closed-loop system (*Figure 2b*). Likewise, online detected events scored higher on metrics for linear spatial-temporal structure (line fit score, Pearson correlation and sequence score; *Figure 2c,d*). These results are consistent with the closed-loop system preferentially detecting replay-containing reference bursts.

For quantification purposes, and absent ground-truth replay labels, we labeled offline reference bursts with arm-biased posteriors ($bias_{max}$ score >3) and trajectory structure (line fit score >0.1) as containing replay content for one of the maze arms (*Table 1*). Across the three sessions, 35.3% of reference bursts were associated with replay (1238/3506; REST — 18.5%, 244/1321; RUN2 — 45.5%, 994/2185), of which 25.8% were classified as putative joint replay events that span more than one maze arm (*Wu and Foster, 2014*).

In both REST and RUN2, single and joint replay events were correctly detected and classified in real-time (*Figure 3a,b*). The system also correctly ignored bursts that did not contain consistent representations of any of the three maze arms (*Figure 3c*). False-positive detections were mainly caused by local matches to the identification criteria in the 30 ms window without significant reference

**Table 1.** Overview of datasets.

| Dataset | Epoch | # bursts | Burst rate [Hz] | # replay bursts | # bursts w/joint content | Mean burst duration [ms] |
|---|---|---|---|---|---|---|
| 1 | REST | 297 | 0.40 | 42 | 15 | 107.2 |
| 1 | RUN2 | 663 | 0.56 | 345 | 85 | 94.4 |
| 2 | REST | 537 | 0.43 | 68b) F | 34 | 101.2 |
| 2 | RUN2 | 808 | 0.64 | 315 | 77 | 93.6 |
| 3 | REST | 487 | 0.38 | 134 | 46 | 102.5 |
| 3 | RUN2 | 714 | 0.59 | 334 | 62 | 95.6 |

DOI: https://doi.org/10.7554/eLife.36275.007

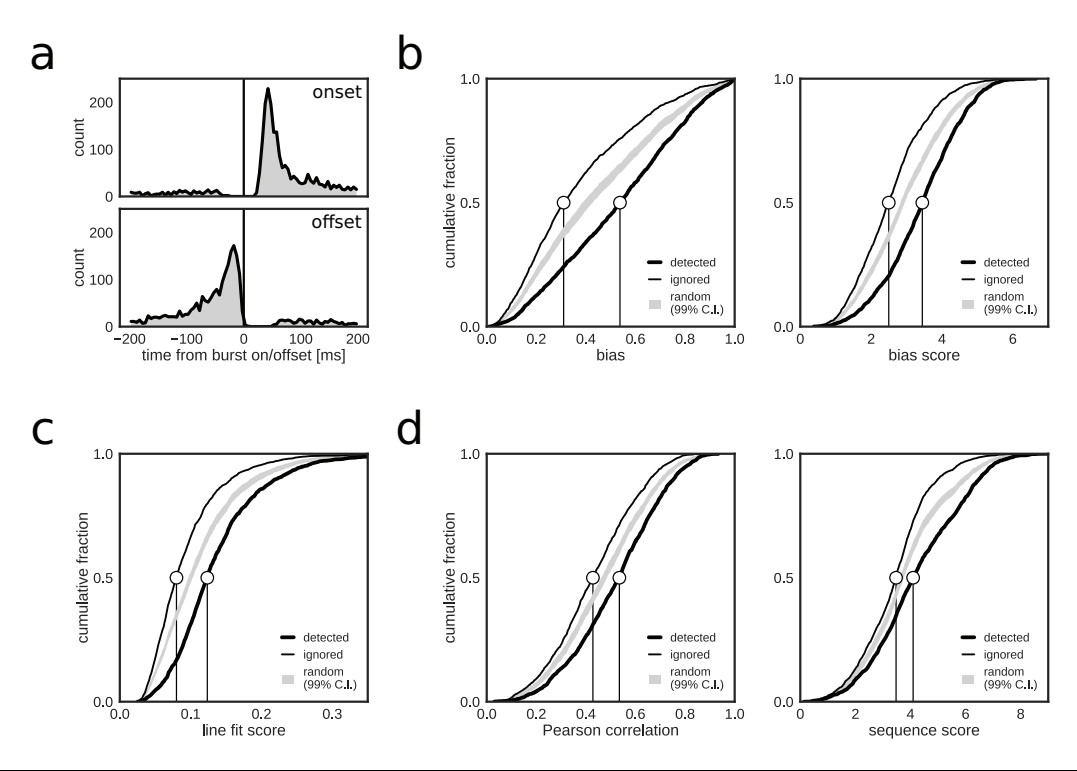

**Figure 2.** Closed-loop detections correlate with replay content. (**a**) Peri-event time histogram of online detections relative to onset (top) and offset (bottom) of reference population bursts. (**b–d**) Cumulative distributions of different replay content measures for reference bursts that were either flagged online as containing replay (thick line) or that did not trigger an online detection (thin line). Gray shaded area shows the 99% confidence interval (CI) of the cumulative distribution for randomized detections of reference bursts (matched to the average online detection rate). Note that reference bursts that were identified online as replay events are associated with higher values for all of the offline replay content measures.
DOI: https://doi.org/10.7554/eLife.36275.008

replay present across the whole population burst (see examples in *Figure 3—figure supplement 1a*). False-negative detections were mainly caused by failure of noisy MUA, sharpness and consistency signals to cross threshold values concurrently (see examples in *Figure 3—figure supplement 1b*). Occasionally, false-positive/-negative detections were the result of a mismatch between the offline and online decoded position estimates, possibly resulting from compression of the encoding model or differences in spike detection.

To quantify the performance, we treated the online detection of replay content in reference bursts as a binary classification problem and determined the number of true/false positives/negatives in the corresponding confusion matrix (*Figure 4a*). Overall, a high fraction of reference bursts

**Table 2.** Replay detection performance I.

| Dataset | Epoch | Out of burst rate [min$^{-1}$] | Content accuracy | Median absolute latency [ms] | Median relative latency [%] |
|---|---|---|---|---|---|
| 1 | REST | 0.24 | 0.95 | 51.77 | 50.85 |
| 1 | RUN2 | 1.83 | 0.99 | 53.77 | 67.54 |
| 2 | REST | 0.33 | 0.96 | 49.77 | 54.10 |
| 2 | RUN2 | 1.05 | 1.00 | 57.39 | 63.81 |
| 3 | REST | 2.37 | 0.83 | 44.62 | 50.49 |
| 3 | RUN2 | 7.46 | 0.87 | 49.95 | 62.53 |

DOI: https://doi.org/10.7554/eLife.36275.013

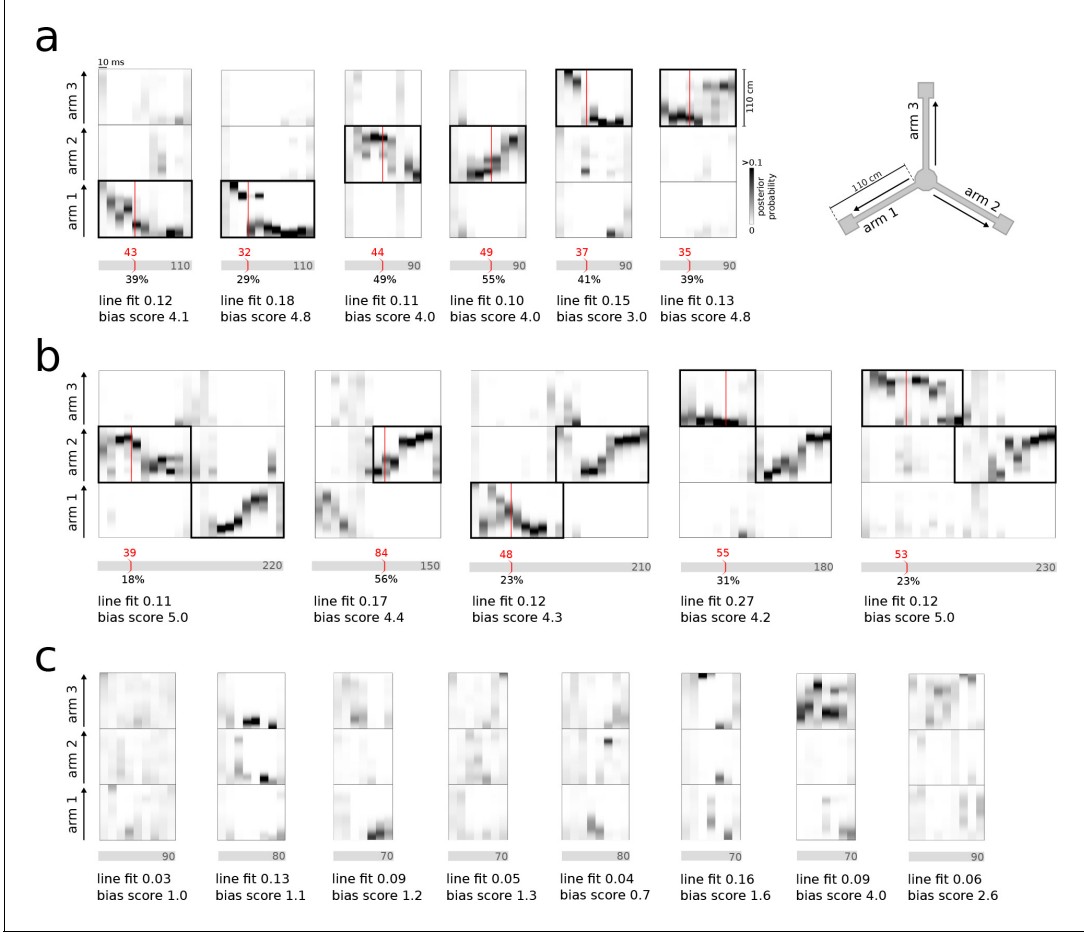

**Figure 3.** Real-time detection of replay content. (a) Examples of reference single content replay events that were successfully detected online with correct identification of replay content (vertical red line). The gray bar below each plot represents the total event with duration (in ms) indicated inside the bar. The absolute detection latency from event onset is indicated in red above the bar and the detection latency relative to the duration of the event is indicated below the bar. (b) Similar to (a) but for joint replay events. All examples except the second show online detections triggered by the first replayed arm; the second example shows an online detection that was triggered by a replay of the second arm. (c) Similar to (a) but for reference bursts without replay content that were correctly ignored online.

DOI: https://doi.org/10.7554/eLife.36275.009

The following figure supplement is available for figure 3:

**Figure supplement 1.** Examples of incorrect online identification of replay.

DOI: https://doi.org/10.7554/eLife.36275.010

with replay content were correctly flagged as such online (median sensitivity 0.78, range: [0.60, 0.95]) and a high fraction of non-replay reference bursts were correctly ignored by the online system (median specificity 0.71, range: [0.44, 0.80]) (*Figure 4b*, top; *Table 3*). The online detection favored a low number of false negatives at the expense of a higher number of false positives (median false omission rate (FOR) 0.11, false discovery rate (FDR) 0.54), in particular during REST (FOR 0.02, FDR 0.68) as compared to RUN2 (FOR 0.28, FDR 0.36) (*Figure 4b*, middle; *Table 3*).

To further characterize the online detection of reference bursts with replay content, we computed unbiased performance measures informedness, markedness and Matthews correlation coefficient (*Table 3*), which are not sensitive to population prevalence and label bias (*Powers, 2011*). All three measures were well above zero, the performance score of a random classifier (*Figure 4b* bottom; *Table 3*), indicating good online identification of population bursts with replay content. For all datasets, the correlation between online hits and offline replay content was highly significant ($\chi^2 > 50$, $p < 2 \times 10^{-12}$).

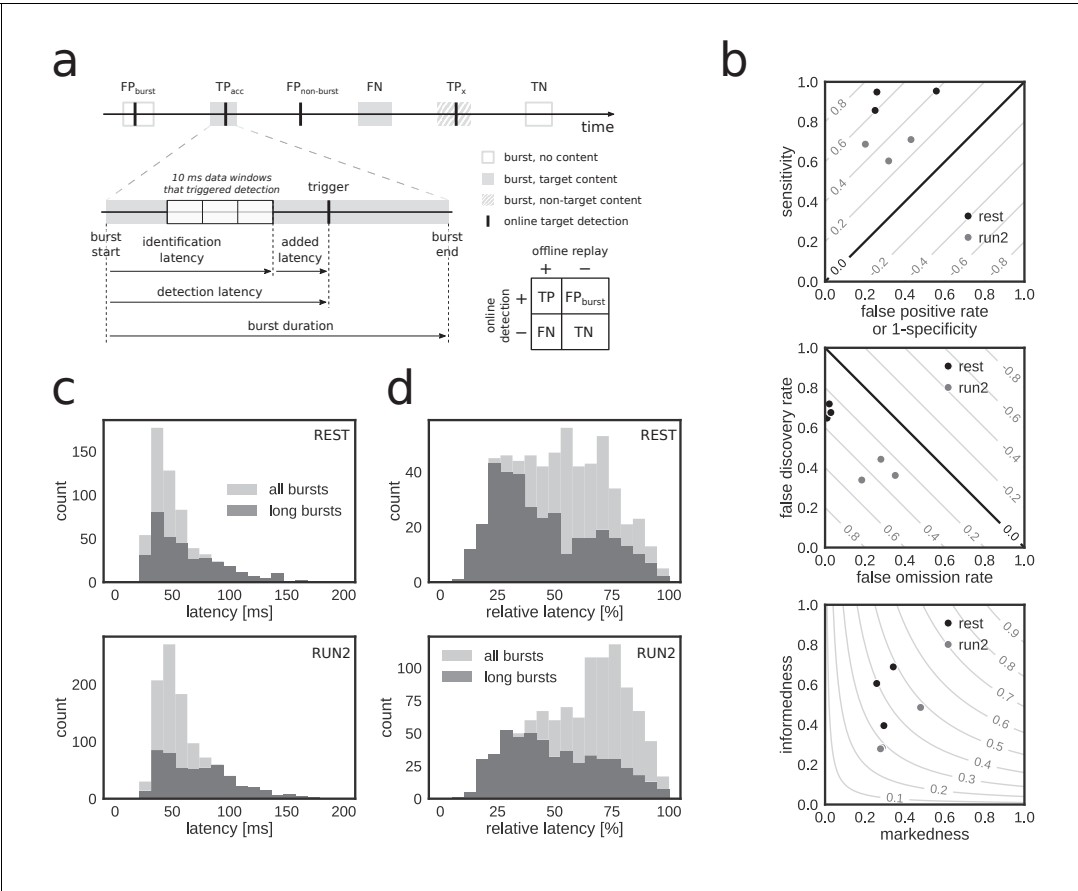

**Figure 4.** Accurate and rapid online identification of replay bursts. (**a**) Schematic overview of the true/false positive/negative classification of online replay detections (top), associated latencies (bottom left) and the confusion matrix used for measuring performance (bottom right). (**b**) Classification performance measures for all REST (black) and RUN2 (gray) test epochs. Diagonal gray lines in the top and middle plots represent isolines for informedness and markedness measures, respectively. Gray lines in the bottom plot represent isolines for Matthews correlation coefficient. In the bottom graph, two RUN2 epoch values overlap in the lowest gray dot. (**c–d**) Distribution of absolute (**c**) and relative (**d**) online detection latency in REST (top) and RUN2 (bottom) epochs. Light gray: all online detected reference bursts. Dark gray: reference bursts with duration longer than 100 ms.
DOI: https://doi.org/10.7554/eLife.36275.011

For the reference bursts that were correctly flagged by the online system as events with replay, the actual replay content (i.e. which of the three maze arms was represented in the replay event) was classified with high accuracy (*Table 2*; median content accuracy 0.95, range [0.83, 1.00]; REST 0.95, RUN2 0.95).

**Table 3.** Replay detection performance II.

| Dataset | Epoch | Sensitivity | Specificity | False omission rate | False discovery rate | Informedness | Markedness | Correlation |
|---|---|---|---|---|---|---|---|---|
| 1 | REST | 0.95 | 0.74 | 0.01 | 0.65 | 0.69 | 0.34 | 0.49 |
| 1 | RUN2 | 0.60 | 0.68 | 0.35 | 0.36 | 0.29 | 0.29 | 0.29 |
| 2 | REST | 0.86 | 0.75 | 0.02 | 0.72 | 0.61 | 0.26 | 0.40 |
| 2 | RUN2 | 0.69 | 0.80 | 0.18 | 0.34 | 0.49 | 0.48 | 0.48 |
| 3 | REST | 0.95 | 0.44 | 0.03 | 0.68 | 0.40 | 0.30 | 0.34 |
| 3 | RUN2 | 0.71 | 0.57 | 0.28 | 0.44 | 0.28 | 0.28 | 0.28 |

DOI: https://doi.org/10.7554/eLife.36275.012

Taken together, these results demonstrate the capability of our closed-loop system to identify and classify hippocampal replay content with high accuracy under realistic experimental conditions during both resting and active animal behavior.

Next, we analyzed the latency of online detection, which is a critical metric for experiments that require on-time closed-loop feedback. Across all reference bursts that were detected online, the median closed-loop detection latency (*Figure 4a*; *Table 2*) relative to the offline-defined burst onset was 50.7 ms (90% CI [31.4, 116.2]) and the median relative detection latency, as percentage of the reference burst duration, was 59.5% (90% CI: [21.9, 89.0]) (*Figure 4c,d*; *Table 2*). The latency for online detection of reference bursts that contain replay content was similar (median detection latency 51.0 ms, 90% CI [30.3, 119.6]; median relative latency 53.6%, 90% CI [19.5, 87.4]). For long reference replay bursts (duration >100 ms), the median detection latency was slightly higher at 58.3 ms (90% CI [30.0, 127.2]), but the median relative latency of 41.9% (90% CI [18.1, 84.7]) was lower (*Figure 4c,d*). These results demonstrate that online replay detection occurred well before burst offset time, thus making our hardware–software system suitable for experiments requiring replay content-specific closed-loop manipulation.

## Effect of parameters on replay content detection

In the experimental tests, the parameters for online replay content identification and classification were chosen on the basis of pilot experiments and offline simulations carried out beforehand. However, such parameters may be tuned to meet the specific requirements of an experiment. For example, it may be desirable to detect the target replay content as completely as possible (i.e. high sensitivity) at the expense of a higher number of false positives (reduced specificity). Conversely, experimenters may want to favor high specificity at the expense of a higher number of missed target replay events.

To characterize the influence of parameters on online detection performance, experimental data were played back offline while applying the online detection algorithm, using a range of values for the MUA and posterior sharpness thresholds $\theta_{mua}$ and $\theta_{sharp}$. As expected, increasing values of $\theta_{mua}$ and $\theta_{sharp}$ parameters reduced the false-positive and false-discovery rates, at the expense of a decrease in sensitivity and an increase in false-omission rate during both REST (*Figure 5—figure supplement 1*) and RUN2 (*Figure 5—figure supplement 2*).

We next asked how close the chosen parameters in the live test were to the pair of optimal parameters that maximize the Matthews correlation coefficient (*Figure 5a*), which balances all four elements of the confusion matrix (true positives (TP), true negatives (TN), in-burst false positives ($FP_{burst}$), and false negatives (FN)). For each optimal pair of parameters, we separately assessed the corresponding rate of non-burst detections ($FP_{non\_burst}$) (*Figure 5b*), which is not accounted for in the maximization of the Matthews correlation coefficient.

We found that we used sub-optimal parameters in REST epochs, suggesting that a better detection could have been achieved by increasing $\theta_{mua}$ and $\theta_{sharp}$ thresholds (*Figure 5 a,b Table 4*). On the other hand, in the RUN2 epochs, the maximum attainable correlations were only marginally and not significantly higher than those obtained in the live tests (*Table 4*; *Figure 5a*, bottom). Moreover, the highest correlation for RUN2 was obtained at the same $\theta_{sharp}$ used in the live tests, but with lower $\theta_{mua}$ thresholds (*Figure 5a*, bottom; *Table 4*), thus allowing for a dramatically lower $FP_{non\_burst}$ rate than the parameters that maximize the Matthews correlation coefficient (*Figure 5b*, bottom; *Table 4*).

Both content accuracy and detection latency were largely unaffected by changes in $\theta_{mua}$ or $\theta_{sharp}$ thresholds (*Figure 5—figure supplement 1e, f* and *Figure 5—figure supplement 2e,f*). Content accuracy is mainly dependent on the online decoding performance and is influenced by compression threshold (see *Figure 1d*) and the number of recorded neurons (or proxies thereof, such as the number of tetrodes). Detection latency is mainly dependent on the size of the integration window used to determine replay content, which was fixed to 30 ms (three 10 ms bins) in the live tests. The simulations show that detection latency varied with the size of the integration window (*Figure 5—figure supplement 3*). As expected, the use of smaller bin size and/or a lower number of $N_{bins}$ decreases relative detection latency, at the expense of reduced detection accuracy (i.e. lower Matthews correlation coefficient). Thus, experimenters may also tune the integration window size to accommodate faster or more accurate online detections.

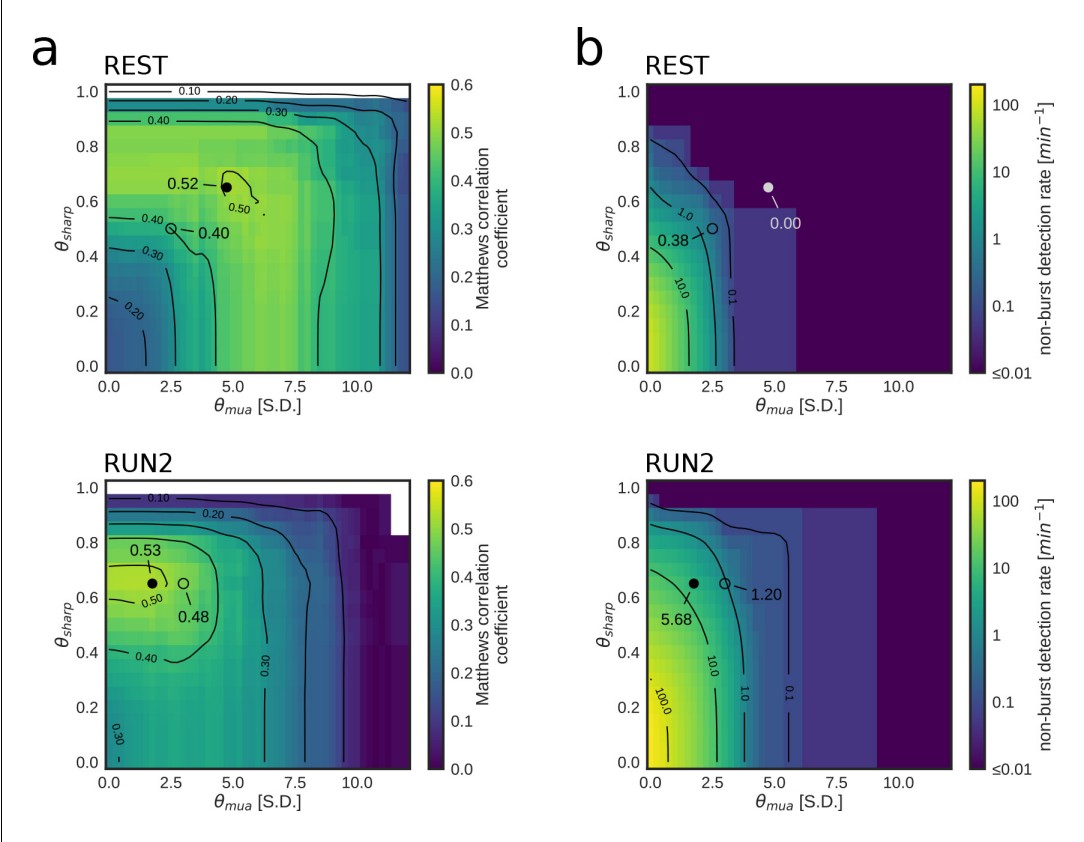

**Figure 5.** Parameter tuning of online replay detection for optimal detection performance. (**a**) Map of the Matthews correlation coefficient for different combinations of values for $\theta_{mua}$ and $\theta_{sharp}$ of a REST (top) and RUN2 (bottom) epoch; the map was computed using offline playback simulations (dataset 2). Circles indicate the value corresponding to the actual parameters used in the online tests (open circle) and the value corresponding to the set of parameters that maximizes the Matthews correlation coefficient (filled circle). (**b**) Same as (**a**) for the non-burst detection rate for different combination of thresholds. Circles indicate the value corresponding to the actual parameters used in the online tests (open circle) and the value corresponding to the set of parameters that maximizes the Matthews correlation coefficient (filled circle).

DOI: https://doi.org/10.7554/eLife.36275.014

The following figure supplements are available for figure 5:

**Figure supplement 1.** Parameter tuning of the replay content identification algorithm for customized detection performance of a REST epoch.
DOI: https://doi.org/10.7554/eLife.36275.015

**Figure supplement 2.** Parameter tuning of the replay content identification algorithm for customized detection performance of a RUN2 epoch.
DOI: https://doi.org/10.7554/eLife.36275.016

**Figure supplement 3.** Replay identification performance as a function of time bin size and number of bins ($N_{bins}$) for both a REST (**a**) and RUN2 (**b**) epoch.
DOI: https://doi.org/10.7554/eLife.36275.017

## Online detection of replay content requires sufficiently high sampling of ensemble activity

Accurate real-time replay detection requires replay content to be predictable from the initial portion of the bursting event. Therefore, higher signal-to-noise and neatly structured decoded replay trajectories are easier to detect online as they are less susceptible to short timescale signal variability. Since the signal-to-noise ratio of replay is strongly influenced by the level of sampling of the spiking neuronal population (although biological factors might play a role too [*Roumis and Frank, 2015*; *Tang et al., 2017*]), spiking data recorded from poorly sampled neuronal populations are likely to lead to an increase in online replay detection errors.

To determine how spike data availability affects online replay identification, we progressively excluded tetrodes in one dataset and performed offline simulations of online replay detection in REST and RUN2 (*Figure 6* and *Figure 6—figure supplement 1*). Consistent with a higher signal-to-

**Table 4.** Optimal parameters.

| Dataset | Epoch | Live $\theta_{mua}$ | Live $\theta_{sharp}$ | Optimal $\theta_{mua}$ | Optimal $\theta_{sharp}$ | Optimal correlation | ΔCorrelation live test[*] | Out of burst rate[min$^{-1}$] |
|---------|-------|---------|-----------|-----------|-----------|---------|---------|---------|
| 1 | REST | 2.50 | 0.50 | 4.00 | 0.80 | 0.76 | +0.27[***] | 0.08 |
| 1 | RUN2 | 3.00 | 0.65 | 0.00 | 0.65 | 0.31 | +0.03 | 28.57 |
| 2 | REST | 2.50 | 0.50 | 4.75 | 0.65 | 0.52 | +0.12[**] | 0.00 |
| 2 | RUN2 | 3.00 | 0.65 | 1.75 | 0.65 | 0.53 | +0.05 | 5.68 |
| 3 | REST | 2.50 | 0.50 | 5.75 | 0.60 | 0.47 | +0.13[**] | 0.05 |
| 3 | RUN2 | 3.00 | 0.65 | 1.00 | 0.65 | 0.33 | +0.05 | 30.81 |

[*]Significance: *=p < 0.05, **=p < 0.01, ***=p < 0.001

DOI: https://doi.org/10.7554/eLife.36275.021

noise ratio, a higher tetrode number decreased the variability of the MUA rate estimate within 10 ms bins and increased the separability of the MUA rate distributions for non-burst periods, bursts without replay and bursts with replay (**Figure 6—figure supplement 2a,c**). The sharpness of the posterior probability distributions increased with higher tetrode numbers, whereas the sharpness distributions for non-burst periods and bursts with replay showed a clear separation (**Figure 6—figure supplement 2b,d**).

To quantify the replay detection performance, a fixed replay reference was used (computed on the original dataset), so that the results were not affected by degradation of the reference. For each simulation, optimal $\theta_{mua}$ and $\theta_{sharp}$ thresholds were computed separately (**Figure 6—figure supplement 1b, d**). As expected, replay identification and classification performance improved when more tetrodes were included, as evidenced by higher correlation and content accuracy measures (**Figure 6a**). Relative detection latency decreased marginally with the number of tetrodes; whereas FP$_{non\_burst}$ rate remained low for REST and was variable for RUN2 (probably because $\theta_{mua}$ and $\theta_{sharp}$ thresholds were not optimized for low out-of-burst detections).

Matthews correlation coefficient does not appear to saturate at the maximum number of tetrodes in the test dataset (14), suggesting that further improvement is possible by increasing the number of sampled cells. Interestingly, Matthews correlation shows little variation for low number of tetrodes and improves most strongly from 10 tetrodes and up. This non-linear dependency is even more clearly present for content accuracy. It is conceivable that, given heterogeneity in spiking data across tetrodes, there is a critical subset of tetrodes that provide most of the information for accurate determination of replay content. The higher Matthews correlation was mainly due to an increased sensitivity and a reduction in false discovery and omission rates (**Figure 6—figure supplement 1a,c**).

Next, we asked to what extent run decoding performance, evaluated through cross-validation prior to closed-loop manipulation, is predictive of good performance of subsequent (online) replay content identification (**Maboudi et al., 2018**; **van der Meer et al., 2017**). As expected, cross-validated run decoding improved with increasing number of tetrodes, as in-arm decoding error decreased and the fraction of arm-correct position estimates increased (**Figure 6b**). Run-decoding performance improved rapidly up to four tetrodes and more gradually for higher tetrode counts. As a result, replay detection performance and run decoding performance show a low and highly non-linear correlation (**Figure 6c**), which suggests that a low run-decoding error (e.g. median error below 10 cm) is a necessary although not sufficient condition for accurate online replay detection.

## Discussion

We demonstrated a solution for real-time detection of hippocampal replay content in freely moving rodents that perform navigation tasks. Our solution provides rapid closed-loop feedback that is specifically triggered by replay of an experimenter-defined content, such as replay of a trajectory through one segment of a maze explored by the animal. This technical advance makes it possible to perform replay content-specific perturbations to address directly the causal link between specific replay events and memory consolidation/retrieval processes.

Our system goes beyond recent efforts to decode or respond to hippocampal activity online. Previous work described systems that provide real-time feedback stimulation in response to spiking

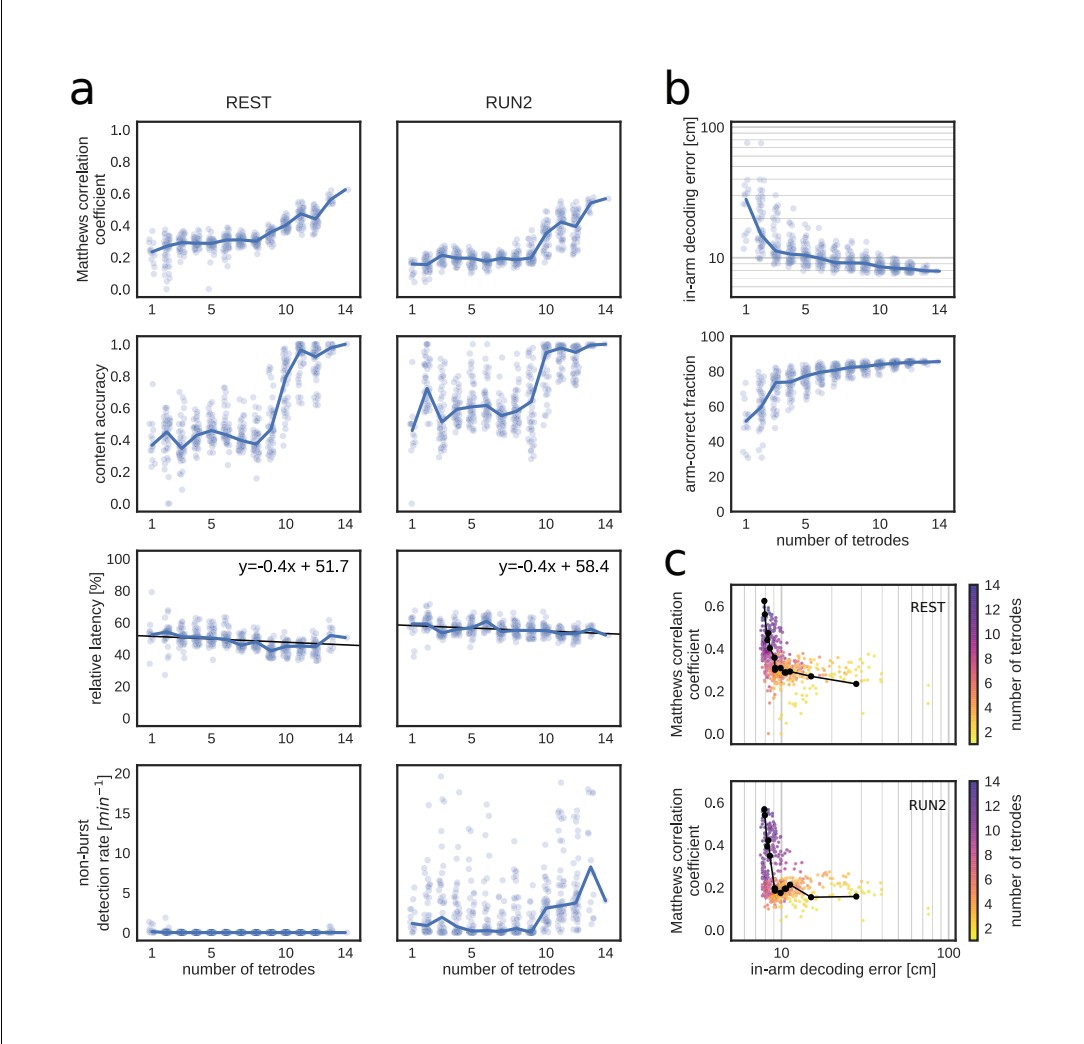

**Figure 6.** Matthews correlation as a measure of replay identification performance as a function of the number of tetrodes included (up to 50 random selections per tetrode number). (a) Matthews correlation coefficient, content accuracy, relative detection latency and non-burst detection rate for REST (left) and RUN2 (right). Dots represent individual tests, lines represent the median. (b) Cross-validated RUN1 decoding performance as a function of the number of tetrodes. Top: median in-arm decoding errors. Bottom: fraction arm-correct position estimates. (c) Scatter plot of the relation between median in-arm decoding error and replay detection performance during REST (top) and RUN2 (bottom). Black lines represent the centroid for each color-coded number of tetrodes in vivo.

DOI: https://doi.org/10.7554/eLife.36275.018

The following figure supplements are available for figure 6:

**Figure supplement 1.** Extended characterization of replay detection performance as a function of number of tetrodes for REST.
DOI: https://doi.org/10.7554/eLife.36275.019

**Figure supplement 2.** Dependence of MUA and sharpness metrics on the number of tetrodes.
DOI: https://doi.org/10.7554/eLife.36275.020

activity (*Nguyen et al., 2014*; *Patel et al., 2017*). These systems perform online decoding of a rat's physical location from a limited number of hippocampal place cells (*Guger et al., 2011*) or implement online spike-sorting with simultaneous detection of ripple oscillations (*Sethi and Kemere, 2015*). However, none of these prior studies addressed the combination of short timescale neural decoding and low-latency detection of replayed spike sequences. *de Lavilléon et al. (2015)* showed that coupling spikes of a single hippocampal place cell during rest to electrical stimulation of the medial forebrain bundle induced a subsequent preference of the animal for the cell's place field. Still, the association with replay was only speculative as no multi-neuronal pattern was being

detected. Finally, *Deng et al. (2016)* developed an algorithm for rapid hippocampal replay content classification in the awake state using state-space models. Despite the low algorithmic latency and the solid mathematical framework of their strategy, the computational latency was still prohibitive and the algorithm's application to live data streams in an experiment was neither demonstrated nor fully conceptualized.

Our replay detection technology has similarities and differences with the large body of research in brain–computer interfaces (BCIs) and neuroprosthetics (*Lebedev and Nicolelis, 2006*). In common with some prototypes of invasive BCIs used for controlling external devices (*Fraser et al., 2009*; *Li and Li, 2017*), we applied online population decoding to unsorted large-scale single-unit recordings. The use of high-level signals that relate to cognitive processes such action planning or memory rather than to pure motor function is not new in neuroprosthetics (see cognitive neural prosthetics (*Andersen et al., 2010*; *Hampson et al., 2018*). In contrast to other BCI and neuroprosthetics applications, the development of a 'replay BCI' faced two unique challenges. First, while it is possible to test the accuracy of BCIs by measuring the difference between desired and actual output, the content of hippocampal replay does not have a 'ground truth' and requires assumptions and statistical analyses to quantify the goodness of the output. The second and most difficult challenge relates to the strict latency specifications in generating an online feedback. BCIs can operate with a processing latency of less than 50 ms when using slowly sampled (below 4800 Hz) EEG signals on few (8–32) channels (*Fischer et al., 2014*; *Wilson et al., 2010*). BCIs using rapidly sampled signals for spike detection on larger electrode arrays can reach latencies as low as 120 ms (*Velliste et al., 2008*); lower latencies are however not necessary as the natural control of motor output has a natural delay of 150 – 200 ms (*Velliste et al., 2008*; *Xu et al., 2014*). Here, we surpassed these limits by using more than 50 channels sampled at 32 kHz to generate estimates of brain state (animals' position) every 10 ms and to classify hippocampal replay patterns with ~50 ms average latency.

As with any system that aims to provide on-time feedback, a trade-off was made between the earliest possible prediction of replay content in a population burst and the most accurate possible prediction. This trade-off is reflected in the numbers of incorrect detections and omissions, when compared to a reference replay content classification. Ultimately, the experimental goals determine how the best trade-off is made. For a balanced trade-off between true and false positives/negatives, we opted for the Matthews correlation coefficient as a measure of performance. For some experiments, however, it may be desirable to be more content-specific and a corresponding increase in false negatives (lower sensitivity) is deemed acceptable. On the other hand, it may be more important to detect as many events with target replay content as possible and an increased rate of false positives (lower specificity) is taken for granted. The two main parameters in our replay content detection algorithm, $\theta_{mua}$ and $\theta_{sharp}$, provide an intuitive means to tune the detection performance and to obtain the desired trade-off. While the exact parameter values depend on the data and optimal values are not known prior to the experiment, similar datasets collected beforehand (without disruptive closed-loop manipulations) are a guide to the actual parameter selection.

The trade-off between accuracy and latency is affected by the performance of neural decoding. A high cell yield in the recordings will improve decoding performance and hence is expected to lead to better online replay detection. The no-spike-sort decoding approach (*Kloosterman et al., 2014*), which is at the core of our solution, helps by utilizing the information from all available spikes, whether or not they are part of well-isolated cell clusters. Compression of the encoding model and software parallelization in Falcon provided the necessary computational speed-up, which in our system results in the availability of the animal's replayed position within 2 ms of data arrival, with acceptable decoding accuracy loss.

A second factor that contributes to the latency-accuracy trade-off is the amount of recent data considered for classifying replay content online. In the content classification algorithm, a shorter window of integration (i.e. fewer bins and/or smaller bin sizes) results in lower detection latencies, and also in an increased focus on local, noisy features of the data that may not be consistent with replay content across the complete event. We settled on a window of 30 ms, which resulted in good replay content detection within short latencies. In future work, further decrease of false-positive and negative detection could be achieved by an extension of our algorithm that adaptively increases the time window to accumulate additional evidence until a certain level of confidence that the event under consideration does or does not represent the target replay content has been reached.

Further improvement of online replay detection is possible by increasing the sampling of spiking data (e.g. by increasing number of recording channels), as the performance in our tests with up to 14 tetrodes did not yet reach a plateau. The increased spiking data will improve both the MUA burst detection and replay content determination components of the online algorithm. This may also allow a further reduction of the integration window, as this is the main determinant of the detection latency. The system as presented will be able to handle the additional computational burden without increasing the computational latency significantly, as it scales to at least 128 recording channels (32 tetrodes) and probably more when run on a 32-core workstation. Scaling to several hundreds or thousands of recording channels in future implementations may require offloading the computationally intensive neural decoding steps to the graphics processing unit (GPU).

The detection of hippocampal replay events has to take into account that ground truth replay labels are unavailable. To evaluate the performance of online replay detection, we showed that detected events score higher on measures that correlate with replay content and that detections largely agree with an offline-defined reference. The quality of the offline reference – that is, how well it captures the desired target replay content – depends on the parameters of the offline detector and on sufficient sampling of spiking data. The offline detector has the advantage of access to all data and of not being bound by limited compute time, whereas the the online detector includes compromises to perform fast and early real-time detection. Hence, one would generally have a higher confidence in the offline reference and would accordingly tune the parameters of the online detector using either prior knowledge or training data. Still, depending on parameters, the offline reference may vary from being too inclusive (e.g. all population bursts are marked as replay) to overly exclusive (e.g. no population bursts are marked as replay) and reliance on an inappropriate reference without careful verification will make experiments that manipulate online detected replay events difficult to interpret. Another cautionary note should be made about using decoding performance during run to predict subsequent online replay detection performance: in our tests, run decoding was less sensitive to data degradation and although low run decoding error may be required for good decoding of replay content, it is not a sufficient condition.

Detection latency of online replay detection was on average ~50 ms. In our previous work, using the same hardware–software system, we demonstrated online burst detection (without replay content identification) at ~40 ms latency (*Ciliberti and Kloosterman, 2017*). Here, by combining burst and replay content detection in the algorithm, on average we added only 10 ms to the time that is needed for a basic population burst detection in the same closed-loop system.

The rate of false-positive detections outside candidate replay population bursts were generally low in rest, but more prevalent when the rat was actively exploring the maze, probably due to a higher mean population firing rate during motion than during rest. In offline analyses, information about the animal's behavior (e.g. running speed) or replay-associated electrophysiological markers (e.g. high-frequency ripple oscillation power) are often used to reduce this category of false positives (*Davidson et al., 2009*). The integration of this additional information in a Falcon processing pipeline would be relative straightforward, but this strategy may increase detection latency and introduce new sources of errors.

The incorporation of our real-time replay detection solution in an experiment requires that an encoding model is built from spiking data recorded during an initial exploration of the maze. At present, the encoding model is constructed manually offline and uploaded into the Falcon software during a brief pause of 5 – 15 min between the end of the initial maze exploration and the start of the real-time replay detection. Future enhancements will enable the construction and validation of the encoding model online rather than offline: besides reducing experimental down-time, this will provide immediate feedback on the decoding performance and will enable content-specific closed-loop manipulation of awake replay events as soon as the quality of the encoding model is sufficient.

In our demonstration, the target replay content for online detection was the representation of one of three maze arms. The system can be trivially extended to environments that are composed of more than three arms (e.g. 8-arm radial mazes or set of linear tracks) and to target joint replay content that covers multiple maze arms in succession. For more strict detection of replay trajectories, a local spatial derivative criterion may be added to the replay classification algorithm. Together with an extension of the encoding model with running direction (*Davidson et al., 2009*), such a trajectory criterion would form the basis of a detector for forward and reverse replay events (*Diba and Buzsáki, 2007*; *Foster and Wilson, 2006*). By altering the encoding model, the system could also be

used to target other types of replay content, including reactivation of immobility-associated place cells (*Yu et al., 2017*), non-spatial task-related replay (*Takahashi, 2015*), sequence reactivation of time cells (*Pastalkova et al., 2008*) or spiking sequences in nested theta/gamma oscillations during active behavior (*Zheng et al., 2016*). Our approach is not limited to hippocampal replay bursts and could be applied to cortical (*Ji and Wilson, 2007*; *Peyrache et al., 2009*) or subcortical (*Lansink et al., 2009*) replay as long as it is possible to define a decodable and detectable content.

In conclusion, we successfully bridged the gap between the basic formulation of online replay identification algorithms and an actual experiment-ready system that addresses the challenges associated with the detection of replay content in live neural data streams. The approach presented here will pave the way for a new generation of closed-loop experiments aimed at elucidating the causal role of hippocampal and non-hippocampal internally generated neural activity in memory processes and cognition.

# Materials and methods

## Neural decoding

We employed a Bayesian neural decoding approach that computes the posterior probability over position $x$ given the unsorted spikes on $K$ tetrodes (*Kloosterman et al., 2014*). For each tetrode k, $A^k = \left\{ a^k_1, \ldots, a^k_{n_k} \right\}$ defines the peak amplitude vectors for a set of $n_k$ spikes in the decoding time bin with duration $\Delta$. Following Bayes' rule, the posterior is then given by:

$$P\left(x|A^{1:K}\right) = \frac{P(A^{1:K}|x)P(x)}{P(A^{1:K})} \tag{1}$$

Here, $P(A^{1:K})$ is a normalizing constant and we use a uniform prior $P(x)$. We assume conditional independence of the activity on each tetrode, and thus the joint likelihood is computed as the product of the individual likelihoods:

$$P\left(A^{1:K}|x\right) = \prod_{k=1}^{K} P\left(A^k|x\right) \tag{2}$$

For each tetrode, the spiking statistics are modeled as a marked temporal Poisson process (where each mark is the vector of spike peak amplitudes) and the likelihood is expressed as (dropping super- and subscripts k on the right side for clarity):

$$P\left(A^k|x\right) = \Delta^n \left[\prod_i^n \lambda\left(a_i, x\right)\right] e^{-\Delta\lambda(x)} \tag{3}$$

We refer to *Equations 3* as the encoding model. The tetrode-specific joint rate function $\lambda(a, x)$ and marginal rate function $\lambda(x)$ can be further expressed as a ratio of spike count ($p$) and position occupancy ($\pi$) probability distributions:

$$\lambda\left(a, x\right) = \mu\frac{p(a, x)}{\pi(x)} \tag{4}$$

$$\lambda\left(x\right) = \mu\frac{p(x)}{\pi(x)} \tag{5}$$

Here, $\mu$ represents the mean firing rate. To compute the likelihoods, the rate functions are evaluated using a compressed kernel density based estimator (*Sodkomkham et al., 2016*) with kernel bandwidths of 8 cm and 30 μV respectively for the spatial and spike amplitude dimensions, and a compression level in the range [1, 2].

## Online replay detection and classification algorithm

To detect and classify hippocampal replay events online in a continuous stream of spiking data from an array of electrodes, we used a two-step approach. First, we decoded spatial information from the

hippocampal spikes for each 10 ms of incoming data as described above. Second, we applied a simple classification strategy to the most recently received T ms of data (i.e. $N_{bins}$ x 10 ms) to determine the occurrence of a target replay event. The algorithm was designed to detect targets that represent replay of a specific trajectory (e.g. maze arm) in the environment. We defined a set of criteria and marked the successful detection of a target replay event when all criteria were met:

1. Burst of activity: the average multi-unit activity rate across all tetrodes in the last T ms is higher than the threshold $\theta_{mua}$. This criterion minimizes the chance of false-positive detection outside population bursts.
2. Decoding fidelity: the sharpness of the posterior distribution, defined as the integrated probability within approximately 14 cm of the maximum-a-posteriori (MAP) estimate, is higher than the threshold $\theta_{sharp}$, both for the most recent decoding time bin and on average for the last $N_{bins}$ decoding time bins.
3. Replay content consistency: the MAP estimates in the last $N_{bins}$ decoding bins all fall on the target trajectory (e.g. arm of a maze).

The online detection and classification approach described above acts on fixed time windows in streaming data without knowledge of the exact start and end times of any replay event (which are only accessible offline). This means that the algorithm is expected to trigger multiple detections for the same replay event. By introducing a lock-out period after each positive detection, these multiple detections can be largely avoided. The use of a lock-out period is also consistent with closed-loop experiments that limit stimulation frequency to 4 – 5 Hz to avoid adverse side effects of over-stimulation (*Girardeau et al., 2009*; *Jadhav et al., 2012*).

## Implementation of online replay detection

A live network stream of sample-by-sample (32 kHz) multi-channel data from a DigiLynx acquisition system (Neuralynx, Bozeman, MT) was fed into a workstation equipped with 256 GB of RAM, 40 MB of smart cache and two 16-core CPUs (Intel Xeon(R) CPU E5-2698 V3 @ 2.30 GHz) that support hardware-based simultaneous multi-threading for a total of 64 virtual cores. The workstation ran the open source real-time processing software Falcon (*Ciliberti and Kloosterman, 2017*), in which we defined a data processing graph that continuously performed neural decoding and replay identification algorithms on the incoming stream of data. The processing graph was composed of seven serial stages, some of which were split into parallel pipelines (one for each tetrode, indicated by * below):

1. Read samples: UDP packets with raw sample data are received from the network stream and parsed into time-stamped multi-channel voltage signals, using a two-sample internal buffer (62.5 µs). Signals are next dispatched to a set of parallel pipelines, according to a predefined selection of tetrodes and channel to tetrode mapping.
2. *Spike filter: the signal on each tetrode channel is filtered using a IIR 4th-order Bessel filter (cut-off frequencies at 600 and 6000 Hz). Digital filtering is implemented using a biquadratic structure for improved numerical stability. No additional buffering is applied at this stage.
3. *Spike detection: spikes are detected in the filtered signals using a threshold-crossing algorithm (threshold set to 60 µV). For each detected spike, a peak-finding algorithm extracts the spike amplitudes from the four channels. Spike detection introduced a 1.25 ms buffer.
4. *Decoding - likelihood: using a pre-loaded encoding model constructed ahead of time from training data, the likelihood (*Equations 3*) is estimated incrementally for each detected spike in 10 ms time bins. By computing the contribution of each spike to the likelihood as soon as it arrives, the extra latency (in addition to the fixed 10 ms latency imposed by the decoding time bin) is kept to a minimum. The marginal spike count distribution $p(x)$ and position occupancy distribution $\pi(x)$ (*Equations 4 and 5*) are fully pre-computed on a user-defined grid of maze locations to limit the number of on-the-fly computations. The joint spike count probability distribution $p(a, x)$ (*Equations 4*) is partially pre-computed (at the expense of using extra memory) and only the contributions of the incoming spikes and their peak amplitudes are evaluated online.
5. Decoding - posterior: the likelihoods computed for all tetrodes are combined to obtain the final posterior distribution in every 10 ms time bin.
6. Replay detection: the replay detection and identification algorithm is applied to the posterior probability distributions and the multi-unit activity in the last $N_{bins}$ time bins. In the case of a positive detection, an event marked with the detected content (or 'unknown' if only the burst

criterion was met) is triggered. This stage adds a variable algorithmic latency dependent on the number of time bins $N_{bins}$ used to classify a replay event.

7. Digital output: in the last stage, internal replay detection events are translated to closed-loop TTL pulses through an isolated Digital Input/Output (DIO) module (USB-4750, Advantech Benelux, Breda, The Netherlands) that communicates with Falcon over universal serial bus (USB 2.0). A lock-out period of 75 ms is here imposed after each detection.

To monitor decoding performance online during active exploration on the maze, an additional processor node in the graph constructed a posterior probability distribution for 200 ms or 250 ms time bins from the output of the decoder stage and generated a stream of MAP estimates.

To characterize the time needed by the system to report a positive detection following the availability of the corresponding source data, we defined added latency as the difference between the first sample timestamp after the $N_{bins}$ of data that triggered an online detection (as logged in Falcon) and the time-stamp of the corresponding detection TTL event as recorded in Cheetah (these time-stamps have a common time base as they both originate in the Digilynx acquisition hardware).

## Evaluation of online replay classification

### Experimental procedures

Experimental procedures were approved by the KU Leuven (Leuven, Belgium) animal ethics committee and are in accordance with the European Council Directive, 2010/63/EU. One male Long Evans rat was chronically implanted with a custom micro-drive array (*Kloosterman et al., 2009*) carrying up to 20 tetrodes. Each tetrode was constructed from four twisted 12 µm diameter polyimide coated nickel-chrome wires (Sandvik, Sweden) and gold-plated to an impedance of 300 kOhm using a nanoZ device (White Matter, Seattle, WA). For the surgical procedure, the rat was anesthetized with 5% isoflurane in an induction chamber and mounted in a stereotaxic frame. Throughout the surgery, body temperature, heart and respiratory rates were monitored and the anesthesia was maintained with 1–2% isoflurane delivered via a respiratory mask. The skull was exposed after an incision in the scalp and a hole centered above hippocampal area CA1 (2.5 mm lateral from the midline and 4 mm posterior to Bregma) was drilled to the size of the tetrode bundle. The implant was fixed to the skull using bone screws and light curable dental cement. A screw above the cerebellum served as animal ground. As part of another experiment, twisted bipolar stainless steel stimulation electrodes (60 µm diameter) were implanted in the ventral hippocampal commissure. During a 3 day post-surgical recovery, pain relief was provided through a daily subcutaneous injection of 1 ml/kg Metacam (Boehringer Ingelheim, Germany, 2 mg/ml).

Recordings were performed relative to a reference electrode in white matter above area CA1. Signals were sampled at 32 kHz, digitized and stored using a 128-channel Digilynx SX system and Cheetah acquisition software (Neuralynx, Bozeman, MT). Spike waveforms were detected by Cheetah on digitally filtered (FIR filter, 600 – 6000 Hz) versions of the recorded signals. Simultaneously with recordings, a duplicate data stream from the Digilynx system was routed to a dedicated workstation for real-time analysis, as described above. The position of the rat was tracked using head-mounted light-emitting diodes and an overhead camera.

Prior to electrode implantation, the rat was kept on a restricted diet to reduce body weight to 85 – 90% of baseline and was trained to run back and forth on a 120 cm long linear track for food rewards at the ends. As part of an unrelated study, the rat was trained to perform a reward-place association task in a radial maze for three weeks after implantation and prior to tests of online replay identification. In three daily sessions, the rat explored a 3-arm maze for 15 – 20 min before (RUN1) and after (RUN2) a short 10 – 20 min rest period (REST). The maze was composed of three 90 cm long and 8 cm wide elevated linear tracks that were connected to a central platform (50 cm diameter) and ended in a distal 20 × 20 cm reward platform. The configuration of the maze (angles between arms and orientation in the room) were changed daily in order to enhance novelty-induced replay activity (*Cheng and Frank, 2008*). In RUN1, the rat could retrieve choco-rice treats at each reward platform, which were replenished every time all three arms had been visited. The rat traversed each arm at least 14 times, providing sufficient data for constructing and validating the encoding model. During REST, the rat was placed in a familiar sleep box that was located in the same room. Following REST, the rat again explored the same maze in RUN2 and was awarded a variable amount of choco-rice treats at the reward platforms (again, to enhance replay activity

[*Ambrose et al., 2016*; *Singer and Frank, 2009*]). In RUN2, the rat traversed each arm at least ten times.

## Construction of encoding model for online replay identification

Before the start of REST, the spiking and position data recorded in RUN1 were used to construct the encoding model that was needed for subsequent online decoding and replay identification in REST and RUN2. A dedicated processing graph in Falcon software was executed in RUN1 to extract spike times and peak amplitudes from the multi-channel data stream, as well as to collect a video tracking data stream from Cheetah software using Neuralynx', NET-based NetCom API and a custom C++ router application. This setup ensured that the encoding model could be built with minimal delays and without interruption of data acquisition in Neuralynx' Cheetah software.

For decoding purposes, we treated the maze arms as three separate linear tracks with position as a 1D variable that measured the distance from the central platform along the center line of the arms. Video tracked (x,y) coordinates of the animal's location were first cleared of tracking errors (e.g. removal of positions outside the maze) and short lasting periods of missing data (e.g. due to tracking errors or occlusions of the LEDs) were interpolated, before projection of 2D coordinates to the center line of the nearest maze arm. Running velocity was computed as the Gaussian smoothed gradient of the 1D position variable (bandwidth of Gaussian kernel set to 0.2 s).

Building the encoding model entails the construction of compressed kernel density estimators (*Sodkomkham et al., 2016*) for the spike count and occupancy probability distributions (*Equations 4 and 5*). These distributions, and hence the posterior distribution, were evaluated on a regular grid of positions in the maze arms with 2.15 cm grid spacing. The kernel bandwidths for the position and spike amplitude variables were set to 8 cm and 30 μV, respectively. We used a compression threshold in the range [1.8, 2] to speed up online computations at the expense of a small reduction in decoding accuracy. Only spike and position data at times when the rat was running at a speed >8.5 cm/s were incorporated into the encoding model.

To test the decoding performance, a cross-validation procedure was used in which run epochs (speed >8.5 cm/s) in RUN1 were equally split into a training and testing set. An encoding model was built using the training data set and evaluated on 200 ms time bins in the testing data set. Online replay detection and identification was only tested in recording sessions with a cross-validated median decoding error lower than 10 cm.

## Offline detection of reference replay events

A set of reference replay events were defined offline for comparison to the online detected events. First, a smoothed histogram (1 ms bins, Gaussian kernel with 15 ms bandwidth) of multi-unit activity (MUA) was constructed using all recorded spikes. Slow non-burst fluctuations were removed from the MUA signal by detrending with an exponentially weighted moving average filter applied forward and backwards (span = 7.5 s (750 samples); corresponding to a half-life of 2.6 s) (*Ciliberti and Kloosterman, 2017*). Bursts were defined as periods in which the standard normalized detrended MUA exceeded 0.5 and had maximum of at least 2.5. Multiple bursts were merged if separated by less than 20 ms.

Next, we performed neural decoding on the spiking activity in 10 ms time bins for every detected population bursts, using the approach described above for online decoding. However, for a higher decoding accuracy (and at the expense of increased computation time), a lower compression level (range [1, 1.3]) was used for constructing the compressed kernel density estimators in the encoding model.

Replay content of each decoded population burst was characterized with the following measures computed from the posterior probability distributions:

1. Bias: we computed for each maze arm separately the bias as the time-averaged posterior probability $\frac{1}{t}\sum_t \sum_x P_t(x \in x_{arm}|A^{1:K})$. The maximum bias is in the range $[\frac{1}{3}, 1]$, with the lower bound representing equal posterior probability across the three maze arms. This measure was linearly transformed to a $bias_{max}$ measure in the range [0, 1]. A null distribution was constructed from the $bias_{max}$ measures of 2000 shuffles created by a random circular shift of positions across the three arms (independently for each time bin). This shuffle largely preserved

the local spatial structure of the posterior distributions, but disrupted the temporal structure. The $bias_{max}$ score was defined as the Z-score of the $bias_{max}$ relative to the null distribution.

2. Line fit: the line fit score was computed as the time-averaged posterior probability within 15 cm of the best-fitting constant-velocity trajectory, as determined using a modified radon transform (*Davidson et al., 2009*; *Kloosterman, 2011*). The line fit score was only defined for the maze arm associated with $bias_{max}$.

3. Correlation coefficient: 2000 trajectories were sampled from the partial posterior distribution corresponding to the arm associated with $bias_{max}$ and the absolute Pearson correlation coefficient $|r|$ between time and location was computed.

4. Sequence score: the Z-score of $|r|$ relative to the null distribution constructed by computing $|r|$ on 2000 shuffled posterior distributions (random circular shift of positions across the three arms, independently for each time bin) (*Grosmark and Buzsáki, 2016*).

Replayed trajectories may span two arms in a multi-arm maze (*Wu and Foster, 2014*). For this reason, the replay content measures were computed both for the complete burst event and also separately for the two partial events that consist of the first and second half of the time bins in the event. When the number of time bins was odd, the central time bin contributed to both partial events. Given that the maze arms have equal lengths, the implicit assumption is that most joint replay events switch arm representation in the middle of the event. Only long duration burst events (>100 ms) were considered as putative joint replay events.

Full and partial burst events were classified as containing replay content if their respective measures met the following criteria: $bias_{max}$ score >3 and line fit score >0.1. For long burst events with putative joint replay, if either of the partial events were classified as containing replay content and the full event had a $bias_{max}$ that was lower than that of the partial events, then the event was split in half and the two partial events were treated separately. In all, we defined three categories of burst events: short and long burst events without replay content, short and long burst events with a single replay content and long burst events that were split in two with the first, second or both partial events being classified as containing replay content.

## Evaluation of detection performance

Online replay content identification events were matched to offline reference population bursts if the time of detection occurred inside the burst window. For the purpose of evaluating replay detection performance, only the first online detection in each reference burst was considered. Reference bursts with confirmed replay content were counted as either true positives (TP) or false negatives (FN), depending on whether an online detection was associated with the burst or not. TP detections were further classified as accurate ($TP_{acc}$) if the online identified content matched the reference replay content (i.e. the specific maze arm).

We further defined two types of false-positive online detections: those that occurred during reference bursts without replay ($FP_{burst}$) and those that occurred outside any reference burst ($FP_{non\_burst}$). True negatives (TN) were defined as reference bursts without replay content that were also correctly ignored online.

Special consideration was given to the putative joint replay events that were split into two partial events. If the first online detection occurred in the first partial event, it was either counted as TP or FP (depending on whether the partial event contained replay content). In this case, the second partial event was not taken into account. If the first online detection occurred in the second partial event, then a TN or FN was counted for the first partial event and a TP or FP was counted for the second partial event (again depending on whether or not the partial events contained replay content). If no online detection occurred, then one FN was counted.

To characterize the online detection and identification performance, the following metrics were computed:

- Sensitivity, which measures the fraction of reference bursts with replay content that were detected as such online and is computed as TP/(TP +FN).
- Specificity, which measures the fraction of reference bursts without replay content that were correctly ignored online and is computed as: TN/(TN +$FP_{burst}$). The false-positive rate is the complement of specificity and measures the fraction of reference bursts without replay content that were incorrectly marked as replay events online.

- False omission rate (FOR), which measures the fraction of online ignored bursts that contain replay content after all and is computed as FN/(FN +TN).
- False discovery rate (FDR), which measures the fraction of online detections that incorrectly marked a reference burst without replay content and is computed as $FP_{burst}$/($FP_{burst}$ +TP).
- Content accuracy, which measures the fraction of the online detections that correctly indicated the presence of replay content were classified accurately (i.e. correct maze arm) and is computed as $TP_{acc}$/TP.
- Informedness (Youden's index), which is an unbiased performance measure that balances sensitivity and specificity, computed as: sensitivity + specificity – 1.
- Markedness, which is an unbiased performance measure that balances false omission and false discovery rates, computed as: 1 – false omission rate – false discovery rate.
- Matthews correlation coefficient (MCC) computed as $\sqrt{informedness \times markedness}$.

For each (first) online detection that occurred inside a reference burst, the absolute detection latency was computed as the time between the start of the reference burst and the time of the TTL event triggered by the online detection. The relative detection latency was computed as the absolute latency divided by the reference burst duration.

To test the statistical significance of the Matthews correlation coefficient, we used the $\chi^2$ test with the following relation between the correlation and $\chi^2$ statistic: $\chi^2 = N \times MCC^2$, where $N$ is the number of samples used to compute the correlation coefficient. To assess the difference between two Matthews correlation coefficients, we applied the Fisher z-transform and compared the difference of the z-scored correlation coefficients to a normal distribution with standard error $\sigma = \sqrt{\frac{1}{N_1-3} + \frac{1}{N_2-3}}$.

## Parameter characterization

During live tests of replay detection, the parameters of the algorithm were fixed to $\theta_{mua}$= 2.5, $\theta_{sharp}$ = 0.5 and $N_{bins}$= 3 for online detection in REST and to $\theta_{mua}$= 3, $\theta_{sharp}$ = 0.65 and $N_{bins}$= 3 for online detections in RUN2. Offline, the effect of varying the threshold parameters $\theta_{mua}$ and $\theta_{sharp}$ on replay detection and identification performance was evaluated for all datasets. A simulator program, written in the Python language, applied the online algorithm (including 75 ms lock-out) to the saved posterior distributions that were produced by the decoder stage in Falcon's processing graph. For the offline simulations, the closed-loop detection latency is unavailable and the identification latency, defined as the difference between the start of the reference burst and the end of the $N_{bins}$ x 10 ms time window that triggered a positive detection, is reported instead.

## Testing the effect of data availability

To explore the effect of data availability on online replay content detection, subsets of tetrodes were repeatedly sampled from one dataset (dataset #2). For each sample, online detection was simulated offline with playback of spiking data (previously recorded with Cheetah software). For each size of the tetrode subset (minimum 1 and maximum 14), we sampled either all possible combinations or 50 randomly selected combinations (whichever was smaller). For all subsampled tetrode sets, replay detection performance was assessed as before with $\theta_{mua}$ and $\theta_{sharp}$ parameters optimized separately each time. In all simulations, we used the reference content for performance evaluation that was computed on the original full dataset as described above. Reported detection latencies exclude the contribution of the online added computational latency.

## Stress tests of closed-loop system

Stress tests were aimed at determining the added latency of the online decoding system in conditions with high spike rates (from 0 to 1000 spikes/s per tetrode) and high tetrode/channel counts (16-tetrode/64-channel, 24-tetrode/96-channel and 32-tetrode/128-channel).

An analog square wave signal was used as artificial spike generator and fed into four 32-channel Signal Mouse testing boards (Neuralynx, Bozeman, MT, USA) that were connected to the analog preamplifiers and DigiLynx acquisition system. After attenuation by the testing boards, the amplitude of the square wave was 5 mV and the frequency was varied from 0 to 1000 Hz.

Falcon software ran a processing graph that was very similar to that used for live experiments (*Figure 1—figure supplement 1a*), with a few modifications. First, file serializer nodes were

removed and a separate dispatcher node for each group of 32 channels was used (to minimize chances of missed packets). Second, to simulate high tetrode numbers, the parallel per-tetrode processing pipelines (including encoding models built from the original data) were duplicated to test up to 32 tetrodes. Third, to replicate the computations performed during live tests, the neural decoding nodes used pre-recorded spike amplitudes instead of the amplitudes of the incoming artificial spikes. Finally, the replay content classifier node was programmed to generate a closed-loop feedback event every 100 ms (10 bins) for 5 min (for a total of 3000 test latencies). Added latency was computed as the difference between the recorded time-stamp of the closed-loop feedback event and the time-stamp that marked the end of a 100 ms test period.

Stress tests were performed on the same 32-core workstation used for live tests (see 'Implementation of online replay detection') and also on a 4-core machine (single CPU, Intel Core i7-4790 @ 3.60 GHz, 16 GB of RAM and 8 MB of smart cache). Both machines used Hyper-Threading and ran a 64-bit Linux Mint 18 (kernel version: 4.4.0_2.1 generic) operating system.

## Additional information

### Funding

| Funder | Grant reference number | Author |
| --- | --- | --- |
| Fonds Wetenschappelijk Onderzoek | G0D7516N | Fabian Kloosterman |

The funders had no role in study design, data collection and interpretation, or the decision to submit the work for publication.

### Author contributions
Davide Ciliberti, Conceptualization, Data curation, Software, Formal analysis, Validation, Visualization, Methodology, Writing—original draft, Writing—review and editing; Frédéric Michon, Investigation, Visualization, Writing—review and editing; Fabian Kloosterman, Conceptualization, Software, Supervision, Funding acquisition, Visualization, Project administration, Writing—review and editing

### Author ORCIDs
Davide Ciliberti (iD) http://orcid.org/0000-0003-1229-642X
Frédéric Michon (iD) http://orcid.org/0000-0003-1289-2133
Fabian Kloosterman (iD) http://orcid.org/0000-0001-6680-9660

### Ethics
Animal experimentation: This study was performed in accordance with the European Council Directive 2010/63/EU on the protection of animals used for scientific purposes. All animals were handled according to protocols approved by the KU Leuven animal ethics committee (project license number: 119/2015).

### Decision letter and Author response
Decision letter https://doi.org/10.7554/eLife.36275.028
Author response https://doi.org/10.7554/eLife.36275.029

## Additional files

### Supplementary files
• Transparent reporting form
DOI: https://doi.org/10.7554/eLife.36275.022

### Data availability
Datasets are available on the Collaborative Research in Computational Neuroscience website (http://crcns.org). The code for online decoding is integrated into the Falcon software platform and

the latest version can be retrieved from https://bitbucket.org/kloostermannerflab/falcon-server (copy archived at https://github.com/elifesciences-publications/falcon-server). The code for offline analysis can be obtained from https://bitbucket.org/kloostermannerflab/ciliberti_elife2018_realti-mereplay (copy archived at https://github.com/elifesciences-publications/Ciliberti_Elife2018_RealTimeReplay).

The following dataset was generated:

| Author(s) | Year | Dataset title | Dataset URL | Database, license, and accessibility information |
|---|---|---|---|---|
| Davide Ciliberti, Frédéric Michon, Fabian Kloosterman | 2018 | Extracellular recordings from rat hippocampal area CA1 during rest and exploration of a 3-arm maze along with results of real-time and offline detection of hippocampal replay content | http://dx.doi.org/10.6080/K0PN93T4 | Publicly available at Collaborative Research in Computational Neuroscience - Data sharing. |

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
