## [Decision Letter]

Thank you for sending your article entitled "Real-time classification of experience-related ensemble spiking patterns for closed-loop applications" for peer review at *eLife*. Your article is being reviewed by Sabine Kastner as the Senior Editor, a Reviewing Editor, and three reviewers.

Given the list of essential revisions, the editors and reviewers invite you to respond with an action plan for the completion of the additional work. We plan to share your responses with the reviewers and then issue a binding recommendation.

All reviewers were generally positive about your manuscript and saw the potential for the real-time decoding methods to permit experiments that are not currently possible. However, the reviewers engaged in a thoughtful discussion and agreed that there are five key issues that should be addressed to ensure that the study makes a substantial impact on the field. These five specific issues are listed below (and followed by the reviews in their entirety, which include other minor comments that should also be addressed prior to resubmission):

1) The authors should better define their parameter space in which their algorithm is effective (e.g., what are the minimum number of tetrodes and spikes per time bin required for effective online replay detection?). To assess this, the authors could subsample their data (or perhaps use others' datasets that are available online) and focus on how these parameters affect detection of false positives and false negatives.

2) The authors should make their dataset available for comparison with other datasets or future datasets. The authors should also make their code for online and offline detection available for comparison with others.

3) The authors should be sure to more clearly explain how the present paper provides valuable insights that were not provided in their previous report (e.g., parameters necessary for effective online detection of replay).

4) The authors should provide a better explanation of the differences between the new real-time algorithm and standard offline methods.

5) The authors should clarify how examples were selected. Are the presented examples representative or are they the best examples? Perhaps a supplemental figure could be added in which the authors show multiple detected replay events selected in an unbiased/less biased way (e.g., all detected replay events from one example episode of awake rest, or all detected replay episodes from all rest epochs from an example behavioral session).

*Reviewer #1:*

In the current manuscript, Ciliberti et al., describe a novel closed-loop method for online detection of the spatial content encoded by ripple events to allow for near-real-time feedback during specific replay events. This method addresses an important technical problem for in vivo hippocampal research, namely that current work on hippocampal sequences is necessarily correlative. To demonstrate causal relationships between ripple-based reactivation (or other sequences, such as theta sequences) and behavior/memory, one must manipulate neural activity only during expression of specific sequences. As the authors note, one prior replay detection algorithm has been published, although the method described in the current manuscript is distinct from that described by Deng et al.

I have very few concerns regarding the methodology described in the manuscript. I think the authors have done an excellent job of quantifying the accuracy of their algorithm with the dataset they collected, and the authors have convinced me that their method is generally effective at identifying specific replay trajectories in one rat. However, I do have concerns regarding the general applicability of this method beyond the single example recording described in this paper.

If the authors were using this methodology to address a specific scientific question, then it would be acceptable to demonstrate how well it works on their dataset and stop there. However, if the authors are attempting to provide this methodology as a general tool to the scientific community, it becomes much more important to establish the parameter space in which it works and in which it fails. The authors do explore some parameter tuning, but this is largely limited to adjusting *θ_sharp_* and *θ_mua_*. It seems to me that many additional parameters necessarily impact both the quality and the speed of this algorithm. While it is impractical for the authors to examine all possible parameters that may ever be encountered, there are two critical parameters that I feel should be examined by the authors to make this method generally useful to the broader scientific community.

The first of these parameters is the total number of spikes that are entering the algorithm per time bin. From my understanding of the Bayesian decoding algorithm, more spikes generally produces a higher maximum posterior probability, which would be expected to result in more *θ_sharp_* crossings. I also presume that more spikes might generally produce a more accurate representation of the current information representation in the hippocampus and thus increase the likelihood of true replay detection. Further, a higher average spike rate would likely serve to lower bin-to-bin spike density fluctuations, thereby reducing spurious *θ_mua_* crossings. Finally, based on Figure 1C, it seems that more spikes slows the decoding algorithm. Thus, I wonder if there is an optimum number of spikes/ms for this method. In other words, would too many recording electrodes actually become detrimental to implementation of this method (by slowing the decoding too much or making the posterior probabilities too high)? Alternatively, are there a critical number of electrodes or spikes/ms that are required for this method to be effective (to avoid too many spurious *θ_mua_* crossings or to ensure faithful decoding)? These questions could be addressed by subsampling the authors' datasets in order to test whether there is a minimum required spike rate. In addition, there are several publicly available datasets that may contain more spikes/ms or the authors could artificially add spikes to their existing dataset to test the effect of increasing total sampling on the algorithm's performance.

The second parameter is the effective 'coverage' of the environment by the spiking activity. While the authors report that there was sufficient coverage in order to faithfully decode the animal's location during exploration (these data should be shown), it was not reported whether the decoding was more/less accurate on specific arms or whether that accuracy impacted the performance of the algorithm in detecting replays of that arm. This would be important information for researchers to have in order to allow them to know when this method can be properly utilized and when it can't. Specifically, is there a critical decoding accuracy (during active behavior using a 200-300 ms decoding window) that must be met before the replay detection algorithm is effective? This could be examined by the authors by selectively subsampling the spikes detected on specific arms to reduce the effectiveness of the encoding model in order to correlate decoding accuracy with replay detection. This additionally addresses how long RUN1 needs to be for the algorithm to work (i.e., are five laps sufficient to build the encoding model or does it require 20 laps?).

These two parameters could also be tested, perhaps more convincingly, by obtaining data from additional rats, although I do not believe that this is strictly necessary.

*Reviewer #2:*

This Tools and Resources article is the first to demonstrate classification of hippocampal replay using real-time ensemble decoding. This is an achievement that anyone with an interest in replay has likely dreamed of for many years, and to see it finally done is cause for celebration. The ability to detect specific replay content with sufficiently short latency to apply closed-loop (Kloost-loop?) interventions such as medial forebrain stimulation, or optogenetic inhibition of a downstream target, is a major step forward that heralds a potentially transformative wave of experiments.

Of course, this report stops short of actually implementing a closed-loop manipulation, but I do not think that diminishes the magnitude and importance of what the authors have accomplished. The key result is that the required computations, notably clusterless ensemble decoding and subsequent classification, can be performed with impressively short latency (<2 ms, Figure 1D) in a fully working system that is streaming data from a single "proof-of-principle" subject.

In combining ensemble decoding and application to replay, this demonstration is a clear advance compared to previous work, although the authors could be more comprehensive in referencing not only the de Lavilleon et al., (2015) paper in the introduction, but also the work by Guger et al., (2011) and that of Bartic, Battaglia and colleagues (Nguyen et al., 2014; Bartic et al., 2016). Similarly, there is an extensive literature on brain-machine interfaces for control of prosthetic limbs and computer cursors, including real-time decoding systems by Nicolelis, Donohue, Schwartz, Andersen and others. The manuscript would benefit from a paragraph clarifying the similarities and differences with that body of work.

The hardware and software components of the system are clearly described and visualized (Figure 1—figure supplement 1 is particularly helpful). The extensive set of offline data analyses are well justified and appropriate, and the authors provide a fair and balanced discussion highlighting areas where the system can be improved.

I have only one major comment, and several minor suggestions, which I expect the authors to be able to address in short order.

I strongly suggest that the authors make the data sets used publicly available. Doing so would greatly facilitate comparison of potential future improvements built by others to the original results and therefore enhance the impact of the work. Relatedly, it was unclear to me from the manuscript whether the described system is currently implemented in the current version of the open-source Falcon software, please clarify.

*Reviewer #3:*

Hippocampal replay is one of the more interesting information-carrying phenomena identified in the brain. Despite more than a decade of study, however, we know less. The authors present a characterization of a real-time replay detection algorithm. Their tool would be broadly useful in experiments in which one wished to understand how the information represented in hippocampal activity during sharp wave ripples impacts learning and memory. The need for these experiments was further brought home to me as I tried to summarize my concerns about how true-positive and false-positive events were selected. Consequently, I am very excited about this work.

The main innovation I find in the paper is developing and characterizing a real-time replay detection algorithm. It builds on the authors' development of a real-time processing architecture (Ciliberti and Kloosterman, 2017). As much as I am sympathetic with the need to elevating the profile of innovation in neuroengineering, it is unclear to me whether this work is appropriate for *eLife*. In particular, there did not appear to be a scientific discovery. I believe that there are hints that there may be something interesting revealed about the hippocampus in the performance of the real-time algorithm, insofar as the presence of a complete replay event and its spatial content may be predictable from some fraction of the initial portion. Clearly if this was a focus of the paper that would meet the standard. Absent such a scientific result, the work might be more appropriate for a more engineering-focused journal.

An alternative scientific result would have been some behavioral effect of disruption of replay. The lack of such an effect, which the presence in the methods of a description of implantation of stimulation electrodes into the ventral hippocampal commissure suggests was the subject of experimental exploration, is striking. Specifically, the authors explore how the parameters of their real-time system affect the detection performance relative to an offline standard. The performance is quite good, but what if this reflects the fact that the offline standard is in appropriate? How the choice of offline-standard parameters affects the relative real-time detection is not explored.

[Editors' note: further revisions were requested prior to acceptance, as described below.]

Thank you for resubmitting your work entitled "Real-time classification of experience-related ensemble spiking patterns for closed-loop applications" for further consideration at *eLife*. Your revised article has been favorably evaluated by Eve Marder (Senior Editor), a Reviewing Editor, and three reviewers.

We are almost ready to accept your manuscript, but there are a few remaining issues that need to be addressed before acceptance, as outlined below:

Specifically, the reviewers would like the figures included in the rebuttal letter to be included as figure supplements or elsewhere. Please follow this suggestion or provide a rebuttal explaining why you prefer not to do so.

We expect the Editors to be able to make a final decision without requiring review.

*Reviewer #1:*

The revised manuscript is much improved, and the authors have largely addressed my concerns.

I would have much preferred for Figure 6 to have examined the effect of spike or cell number rather than tetrode number, as one can put in 100 tetrodes and get no cells or put in 12 tetrodes and get 150 cells (Wilson and McNaughton, 1993). However, I think the figure included in the response (Figure A) addresses this concern and should be included as Figure 1—figure supplement 4.

I appreciate the analyses included in Author response image 2 in the response, and I think it is valuable information for someone planning to use this method, but I don't consider it strictly necessary (given the relatively modest improvement in accuracy beyond 1 lap). Considering that there are no limits (as far as I know) for supplemental data, and considering that the figure is already made, I see no reason not to include it.

*Reviewer #2:*

In this revision, the authors have added major new analyses to what was already an impressive demonstration of the first real-time ensemble decoding system for hippocampal replay content. The new analyses are important in delineating the tradeoffs involved in configuring the system, and the conditions that need to be met for adequate performance.

Among these additions is the subsampling analysis which suggests a nonlinear performance improvement around 10 tetrodes, and continued improvement as more tetrodes are added. Also useful is the analysis describing how much sampling of the environment is required to build a usable encoding model (Author response image 2 in the rebuttal). If the authors prefer to not include this figure as a supplement, I think a statement in the text along with some descriptive statistics would be helpful in making the point that for this data set, a few trials appear to be sufficient.

My other comments, such as the suggestion to make the data freely available, have also been satisfactorily addressed.

*Reviewer #3):*

The authors additions have satisfied my concerns.

---

## [Author Response]

[Editors' note: the authors’ plan for revisions was approved and the authors made a formal revised submission.]

All reviewers were generally positive about your manuscript and saw the potential for the real-time decoding methods to permit experiments that are not currently possible. However, the reviewers engaged in a thoughtful discussion and agreed that there are five key issues that should be addressed to ensure that the study makes a substantial impact on the field. These five specific issues are listed below (and followed by the reviews in their entirety, which include other minor comments that should also be addressed prior to resubmission):1) The authors should better define their parameter space in which their algorithm is effective (e.g., what are the minimum number of tetrodes and spikes per time bin required for effective online replay detection?). To assess this, the authors could subsample their data (or perhaps use others' datasets that are available online) and focus on how these parameters affect detection of false positives and false negatives.

As the reviewers requested and as detailed below in the replies, we have added analyses in which we subsampled the number of tetrodes and assessed the effect on replay detection performance. We further performed stress-tests of our system to evaluate how many spikes/bin and how many tetrodes can still be processed in real-time without increasing detection latency. Finally, we analyzed how two additional parameters (time bin size and number of bins) influence the trade-off between replay detection latency and accuracy.

2) The authors should make their dataset available for comparison with other datasets or future datasets. The authors should also make their code for online and offline detection available for comparison with others.

We are preparing the three the datasets that we collected for the manuscript for uploading to the “Collaborative Research in Computational Neuroscience” website (http://crcns.org). The code for online decoding is integrated into our Falcon software platform and the latest version will be made available on https://bitbucket.org/kloostermannerflab/falcon-server. The code for offline replay detection (and other analysis) will also be made available in a repository on our bitbucket website (https://bitbucket.org/kloostermannerflab). We are currently in the process of checking and cleaning the repositories and will make them public within the next weeks.

3) The authors should be sure to more clearly explain how the present paper provides valuable insights that were not provided in their previous report (e.g., parameters necessary for effective online detection of replay).

In our previously published work, we described an efficient offline implementation of spike-sorting-less neural decoding (Sodkomkham et al., 2016) and a general software framework for complex online neural processing in closed-loop settings (Ciliberti and Kloosterman, 2017). However, we did not previously address the full real-time replay identification problem with a concrete implementation that could be applied to streaming data. In the current work, we demonstrate that replay events can be identified in-time with a relatively straightforward algorithm and we show how the value of a number of critical parameters influence online replay detection performance. We have more clearly highlighted the novel contributions of the current manuscript in the Introduction.

4) The authors should provide a better explanation of the differences between the new real-time algorithm and standard offline methods.

The main difference is that the online algorithm is designed with real-time streaming data in mind and the complexity is low on purpose such that parameter values are intuitive, which facilitates incorporation into experiments for which parameters need to be set ahead of time and cannot be optimized afterwards. Offline methods have an advantage in that all data is available and there is little computational constraint. We have added a more detailed explanation of the differences between the online and offline methods in the Discussion section.

5) The authors should clarify how examples were selected. Are the presented examples representative or are they the best examples? Perhaps a supplemental figure could be added in which the authors show multiple detected replay events selected in an unbiased/less biased way (e.g., all detected replay events from one example episode of awake rest, or all detected replay episodes from all rest epochs from an example behavioral session).

The examples were picked manually from the first 300 candidate replay events among all RUN2 and SLEEP sessions. The selected examples are not “the best” according to some metric, but certainly there is a bias because of the visual inspection. Selected examples were focused on showing relatively clear replay events with correct online detections. True negative events were selected randomly. We have added a description of how example selection was performed to the figure legend. Given the large number of events (see Table 1), it will be difficult to show all events in a dataset as suggested. However, as was suggested in a follow-up to this reviewer report, we have added a supplemental figure with examples of false positive/negative detections and describe in the legend the main causes of the incorrect detections.

Reviewer #1:

*In the current manuscript, Ciliberti et al., describe a novel closed-loop method for online detection of the spatial content encoded by ripple events to allow for near-real-time feedback during specific replay events. This method addresses an important technical problem for* in vivo *hippocampal research, namely that current work on hippocampal sequences is necessarily correlative. To demonstrate causal relationships between ripple-based reactivation (or other sequences, such as theta sequences) and behavior/memory, one must manipulate neural activity only during expression of specific sequences. As the authors note, one prior replay detection algorithm has been published, although the method described in the current manuscript is distinct from that described by Deng et al.*

Before discussing the points below, we want to highlight that our algorithm is more complete than the on presented in Deng et al., 2016, as it is suitable for an online scenario with streaming data rather than on offline scenario in which the onset and offset of replay events are known. In addition – and in contrast to Deng et al., – we have demonstrated and characterized a real-time implementation.

I have very few concerns regarding the methodology described in the manuscript. I think the authors have done an excellent job of quantifying the accuracy of their algorithm with the dataset they collected, and the authors have convinced me that their method is generally effective at identifying specific replay trajectories in one rat. However, I do have concerns regarding the general applicability of this method beyond the single example recording described in this paper.If the authors were using this methodology to address a specific scientific question, then it would be acceptable to demonstrate how well it works on their dataset and stop there. However, if the authors are attempting to provide this methodology as a general tool to the scientific community, it becomes much more important to establish the parameter space in which it works and in which it fails. The authors do explore some parameter tuning, but this is largely limited to adjusting θsharp and θmua. It seems to me that many additional parameters necessarily impact both the quality and the speed of this algorithm. While it is impractical for the authors to examine all possible parameters that may ever be encountered, there are two critical parameters that I feel should be examined by the authors to make this method generally useful to the broader scientific community. The first of these parameters is the total number of spikes that are entering the algorithm per time bin. From my understanding of the Bayesian decoding algorithm, more spikes generally produces a higher maximum posterior probability, which would be expected to result in more θsharp crossings. I also presume that more spikes might generally produce a more accurate representation of the current information representation in the hippocampus and thus increase the likelihood of true replay detection. Further, a higher average spike rate would likely serve to lower bin-to-bin spike density fluctuations, thereby reducing spurious θmua crossings. Finally, based on Figure 1C, it seems that more spikes slows the decoding algorithm. Thus, I wonder if there is an optimum number of spikes/ms for this method. In other words, would too many recording electrodes actually become detrimental to implementation of this method (by slowing the decoding too much or making the posterior probabilities too high)? Alternatively, are there a critical number of electrodes or spikes/ms that are required for this method to be effective (to avoid too many spurious θmua crossings or to ensure faithful decoding)? These questions could be addressed by subsampling the authors' datasets in order to test whether there is a minimum required spike rate. In addition, there are several publicly available datasets that may contain more spikes/ms or the authors could artificially add spikes to their existing dataset to test the effect of increasing total sampling on the algorithm's performance.

The reviewer is correct that a higher number of spikes (or more accurately, a higher number of informative spikes) improves decoding accuracy with a higher maximum posterior probability. At the same time, more spikes give a better estimate of instantaneous MUA rate. The downside of more spikes is an increase of the computational burden that may affect latency.

To address the reviewer's comment, we first performed a stress test to show what spike rate can be handled in our system with varying number of tetrodes. We performed the test on both a 32-core workstation and a more standard quad-core machine. The results (Figure 1—figure supplement 2) indicate that the added latency remains low even for high spike rates. The quad-core computer generally performs well but has increased median and worst case added latency for high tetrode counts and high spike rates.

Second, we followed the reviewer's suggestion and characterized the replay detection algorithm and its performance when less data is available (by subsampling the tetrodes). As shown in Author response image 1, the reviewer predicted correctly that higher tetrode number decreases the variability and hence increase the separability of the MUA rate distributions for non-burst periods, bursts without replay and bursts with replay. The reviewer was also correct that the maximum posterior probability (as measured by our sharpness measure) shifts to higher values with increasing number of tetrodes, without reaching a plateau in our dataset (panels b and d in Author response image 1). Still, sharpness distributions for non-burst periods and bursts with replay remain separable.

The quantification of the online replay detection performance after tetrode subsampling are described in a new section in the main text and shown in Figure 6 and its supplements. In brief, online replay detection performance increases with higher number of tetrodes, but the increase is strongest when the number of tetrodes is ten or more in our dataset. This suggests that a critical number of tetrodes (and a critical number of informative spikes on these tetrodes) are required for accurate replay detection. In contrast, run decoding error (in 200 ms time bins) during exploration appeared to be less sensitive to reduction of the number of tetrodes. As a consequence, although there is a (highly non-linear) correlation between the variables, a low run decoding error is not a sufficient condition for subsequent good online replay detection.

(We can include the Author response image 1 in the manuscript if the reviewer considers this desirable).

**Author response image 1. respfig1:** (**a**) Example distributions of MUA rate for time bins outside and inside (with or without identified replay content) candidate replay bursts. Rest epoch. (**b**) Mean MUA rate as a function of the number of tetrodes included. Note that mean rate remains constant, but the variance increases with fewer tetrodes. (**c**) Hellinger distance between MUA rate distributions as a function of the number of tetrodes included. (**d**) Optimal *θ_mua_* threshold as a function of the number of tetrodes included. (**a**) Example distributions of sharpness values for time bins outside and inside (with or without identified replay content) candidate replay bursts. Rest epoch. Note shift towards higher sharpness values when more tetrodes are included. (**b**) Mean sharpness value as a function of the number of tetrodes included. (**c**) Hellinger distance between sharpness distributions as a function of the number of tetrodes included. (**d**) Optimal *θ_sharp_* threshold as a function of the number of tetrodes included.

The second parameter is the effective 'coverage' of the environment by the spiking activity. While the authors report that there was sufficient coverage in order to faithfully decode the animal's location during exploration (these data should be shown), it was not reported whether the decoding was more/less accurate on specific arms or whether that accuracy impacted the performance of the algorithm in detecting replays of that arm. This would be important information for researchers to have in order to allow them to know when this method can be properly utilized and when it can't. Specifically, is there a critical decoding accuracy (during active behavior using a 200-300 ms decoding window) that must be met before the replay detection algorithm is effective? This could be examined by the authors by selectively subsampling the spikes detected on specific arms to reduce the effectiveness of the encoding model in order to correlate decoding accuracy with replay detection. This additionally addresses how long RUN1 needs to be for the algorithm to work (i.e., are five laps sufficient to build the encoding model or does it require 20 laps?).These two parameters could also be tested, perhaps more convincingly, by obtaining data from additional rats, although I do not believe that this is strictly necessary.

We have added quantification of the decoding performance during exploration for both RUN1 and RUN2 epochs in Figure 1—figure supplement 3.

To look at the relation between decoding accuracy (during exploration) and online replay detection performance, we subsampled tetrodes and simulated the online replay detection on the downgraded dataset. (To keep the analysis manageable, we opted to subsample tetrodes rather than subsampling of spikes detected on specific arms.) The results are shown in the new Figure 6. As also mentioned above, a low decoding error during exploration does not automatically mean a good online replay detection.

To look at how much data in RUN1 is needed to build a decent encoding model, we built the model with varying training times (Author response image 2). For each training time, we also counted the number of laps that the rat completed for each maze arm. As expected, decoding performance improves with more training time, and a handful of laps are sufficient to reach a relatively stable performance. The figure could be added as a supplemental figure if the reviewer requests this.

**Author response image 2. respfig2:** Decoding performance during exploration in all three datasets as a function of the training time and corresponding number of laps used to build the encoding model. (**a**) Median in-arm decoding error. (**b**) Fraction of position estimates that were located in the correct arm.

Reviewer #2:This Tools and Resources article is the first to demonstrate classification of hippocampal replay using real-time ensemble decoding. This is an achievement that anyone with an interest in replay has likely dreamed of for many years, and to see it finally done is cause for celebration. The ability to detect specific replay content with sufficiently short latency to apply closed-loop (Kloost-loop?) interventions such as medial forebrain stimulation, or optogenetic inhibition of a downstream target, is a major step forward that heralds a potentially transformative wave of experiments.Of course, this report stops short of actually implementing a closed-loop manipulation, but I do not think that diminishes the magnitude and importance of what the authors have accomplished. The key result is that the required computations, notably clusterless ensemble decoding and subsequent classification, can be performed with impressively short latency (<2 ms, Figure 1D) in a fully working system that is streaming data from a single "proof-of-principle" subject.In combining ensemble decoding and application to replay, this demonstration is a clear advance compared to previous work, although the authors could be more comprehensive in referencing not only the de Lavilleon et al., (2015) paper in the introduction, but also the work by Guger et al., (2011) and that of Bartic, Battaglia and colleagues (Nguyen et al., 2014; Bartic et al., 2016). Similarly, there is an extensive literature on brain-machine interfaces for control of prosthetic limbs and computer cursors, including real-time decoding systems by Nicolelis, Donohue, Schwartz, Andersen and others. The manuscript would benefit from a paragraph clarifying the similarities and differences with that body of work.

Thank you for your suggestions. We have updated the discussion to include references to the work of Guger et al., and Nguyen et al. We have also added paragraph describing similarities and differences with brain-machine interfaces and neuroprosthetics and we describe our new method in the wider context of BCIs in the Introduction. We hope these changes address the reviewer’s comments.

The hardware and software components of the system are clearly described and visualized (Figure 1—figure supplement 1 is particularly helpful). The extensive set of offline data analyses are well justified and appropriate, and the authors provide a fair and balanced discussion highlighting areas where the system can be improved.I have only one major comment, which I expect the authors to be able to address in short order.I strongly suggest that the authors make the data sets used publicly available. Doing so would greatly facilitate comparison of potential future improvements built by others to the original results and therefore enhance the impact of the work. Relatedly, it was unclear to me from the manuscript whether the described system is currently implemented in the current version of the open-source Falcon software, please clarify.

Thank you for your suggestion. As mentioned in our reply to the editor’s comments, we are preparing the three datasets for release and will upload them to the “Collaborative Research in Computational Neuroscience” website (http://crcns.org). We are also in the process of adding the processor for decoding and replay detection to the public Falcon repository, and we are preparing the repositories with Python code used to perform offline analysis for public release.

Reviewer #3:Hippocampal replay is one of the more interesting information-carrying phenomena identified in the brain. Despite more than a decade of study, however, we know less. The authors present a characterization of a real-time replay detection algorithm. Their tool would be broadly useful in experiments in which one wished to understand how the information represented in hippocampal activity during sharp wave ripples impacts learning and memory. The need for these experiments was further brought home to me as I tried to summarize my concerns about how true-positive and false-positive events were selected. Consequently, I am very excited about this work.The main innovation I find in the paper is developing and characterizing a real-time replay detection algorithm. It builds on the authors' development of a real-time processing architecture (Ciliberti and Kloosterman, 2017). As much as I am sympathetic with the need to elevating the profile of innovation in neuroengineering, it is unclear to me whether this work is appropriate for eLife. In particular, there did not appear to be a scientific discovery. I believe that there are hints that there may be something interesting revealed about the hippocampus in the performance of the real-time algorithm, insofar as the presence of a complete replay event and its spatial content may be predictable from some fraction of the initial portion. Clearly if this was a focus of the paper that would meet the standard. Absent such a scientific result, the work might be more appropriate for a more engineering-focused journal.An alternative scientific result would have been some behavioral effect of disruption of replay. The lack of such an effect, which the presence in the methods of a description of implantation of stimulation electrodes into the ventral hippocampal commissure suggests was the subject of experimental exploration, is striking. Specifically, the authors explore how the parameters of their real-time system affect the detection performance relative to an offline standard. The performance is quite good, but what if this reflects the fact that the offline standard is in appropriate? How the choice of offline-standard parameters affects the relative real-time detection is not explored.

The reviewer is correct that the long-term goal of this work is to assess the behavioral consequences of content-specific replay disruption. Given the complexity of these experiments, we believe opening the technology to the larger experimental neuroscience community sooner (as a Tools and Resources article in *eLife* and by sharing the tools) provides the best way forward.

We are happy to hear that the reviewer considers the performance of our system “quite good”. The reviewer wonders if the performance level may be due to an inappropriate offline defined standard to which the online detections are compared. This comment touches on an important issue. Although the reviewer is concerned about an inflated online performance, the opposite may be true as well: an inappropriate offline standard that underestimates the online replay detection performance. Fundamentally, the problem is that no ground truth replay labels exist. For this reason, we sought to first characterize the online detection without a hard classification of events into bursts with or without replay content. As shown in Figure 2, positively detected events score higher in several measures that correlate with replay content.

For more in depth quantification of the performance, we did compare the online detections to an offline standard. In our view, an experimenter needs to first define what constitutes replay content and then constructs offline and online detectors that aim to identify the (unlabeled) replay events. The offline detector has access to all data and is not bound by limited compute time and the online detector is (generally) modeled after the offline detector but includes compromises to perform fast and early real-time detection. Hence, we have more confidence in the ability of the offline detector to correctly detect replay events and to provide the best possible reference for evaluation of the online detector performance. One would then try to adjust the parameters for the online detector (e.g. using prior knowledge or training data) to match its result to that of the offline detector.

If the offline standard is “inappropriate” with respect to the (unknown) ground truth (e.g. parameters of the offline detector are set incorrectly), then any (mis)match between the result of the online detector and the offline standard will be difficult to interpret. By changing the parameters of the offline detector, the resulting offline standard may vary all the way from being very inclusive (e.g. all population bursts are marked as replay) to very exclusive (e.g. no population bursts are marked as replay) – and the online detector will largely follow as parameters are adapted (to the extent that the offline and online algorithms are consistent). However, because ground truth is unknown, there is no direct way in which we can assess if the offline standard is appropriate. Indirect methods, such as visual inspection or independent replay assessment (e.g. separate replay measures that are not part of the offline detector), may be helpful and strengthen our belief that the offline standard is appropriate. In our manuscript, visual inspection of the offline defined replay events, as illustrated by the example (non)replay events shown in Figure 3 and the new Figure 3—figure supplement 1, give us confidence that the offline standard is appropriate and contains events with the desired replay structure. We have added some of the consideration above to the Discussion section.

[Editors' note: further revisions were requested prior to acceptance, as described below.]

Reviewer #1:The revised manuscript is much improved, and the authors have largely addressed my concerns.I would have much preferred for Figure 6 to have examined the effect of spike or cell number rather than tetrode number, as one can put in 100 tetrodes and get no cells or put in 12 tetrodes and get 150 cells (Wilson and McNaughton, 1993). However, I think the figure included in the response (Figure A) addresses this concern and should be included as Figure 1—figure supplement 4.

We have added Author response image 1 from the previous reply as supplementary figure to the manuscript. However, given that the data presented relates to the tetrode subsampling analysis, we have included the figure as Figure 6—figure supplement 2.

I appreciate the analyses included in Author response image 2 in the response, and I think it is valuable information for someone planning to use this method, but I don't consider it strictly necessary (given the relatively modest improvement in accuracy beyond 1 lap). Considering that there are no limits (as far as I know) for supplemental data, and considering that the figure is already made, I see no reason not to include it.

We have added the figure as Figure 1—figure supplement 4.

Reviewer #2:In this revision, the authors have added major new analyses to what was already an impressive demonstration of the first real-time ensemble decoding system for hippocampal replay content. The new analyses are important in delineating the tradeoffs involved in configuring the system, and the conditions that need to be met for adequate performance.Among these additions is the subsampling analysis which suggests a nonlinear performance improvement around 10 tetrodes, and continued improvement as more tetrodes are added. Also useful is the analysis describing how much sampling of the environment is required to build a usable encoding model (Author response image 2 in the rebuttal). If the authors prefer to not include this figure as a supplement, I think a statement in the text along with some descriptive statistics would be helpful in making the point that for this data set, a few trials appear to be sufficient.

We have added the figure as Figure 1—figure supplement 4.